# Hashing with Uncertainty Quantification via Sampling-based Hypothesis Testing

**Yucheng Wang** *wangyucheng@tamu.edu*
*Department of Electrical and Computer Engineering, Texas A&M University*

**Mingyuan Zhou** *mingyuan.zhou@mccombs.utexas.edu*
*McCombs School of Business, The University of Texas at Austin*
*Department of Statistics and Data Sciences, The University of Texas at Austin*

**Xiaoning Qian** *xqian@tamu.edu*
*Department of Electrical and Computer Engineering, Texas A&M University*
*Department of Computer Science Engineering, Texas A&M University*
*Computational Science Initiative, Brookhaven National Laboratory*

**Reviewed on OpenReview:** *https://openreview.net/forum?id=cc4v6v310f*

## Abstract

To quantify different types of uncertainty when deriving hash-codes for image retrieval, we develop a probabilistic hashing model (ProbHash). Sampling-based hypothesis testing is then derived for hashing with uncertainty quantification (HashUQ) in ProbHash to improve the granularity of hashing-based retrieval by prioritizing the data with confident hash-codes. HashUQ can drastically improve the retrieval performance without sacrificing computational efficiency. For efficient deployment of HashUQ in real-world applications, we discretize the quantified uncertainty to reduce the potential storage overhead. Experimental results show that our HashUQ can achieve state-of-the-art retrieval performance on three image datasets. Ablation experiments on model hyperparameters, different model components, and effects of UQ are also provided with performance comparisons. Our code is available at https://github.com/QianLab/HashUQ.

## 1 Introduction

Representation learning summarizes high-dimensional data into low-dimensional feature vectors with the key information preserved for conducting various downstream tasks, which has been successfully applied to different domains such as natural language processing (NLP) (Bahdanau et al., 2014), generative modeling (van den Oord et al., 2017; Liu et al., 2019; 2020), data compression (Ballé et al., 2016; Theis et al., 2017; Ballé et al., 2018), and multi-modal machine learning (Chen et al., 2020). Information retrieval can leverage representation learning to efficiently return the most similar data samples within a database given a query. With the power of such representation models, Learning-to-Hash (L2H) (Weiss et al., 2008; Salakhutdinov & Hinton, 2009; Strecha et al., 2011; Liu et al., 2011; 2014; Li et al., 2016) has become one commonly adopted group of algorithms, which are designed for efficient information retrieval based on derived discrete representations in hashing codes (hash-codes). With the extremely compact binary hash-codes derived from high-dimensional data, L2H enables fast comparison and search by comparing the Hamming distance of hash-codes. More recently, the retrieval accuracy of L2H algorithms with different types of data has experienced remarkable improvement, thanks to the rapidly evolving modern computer hardware and advanced deep neural network (DNN) architectures (Krizhevsky et al., 2012; Simonyan & Zisserman, 2014; He et al., 2016; Vaswani et al., 2017) with efficient training algorithms (Hinton et al., 2012; Kingma & Ba, 2014; Bengio et al., 2013; Jang et al., 2016; Maddison et al., 2016; Yin & Zhou, 2019; Yin et al., 2020).

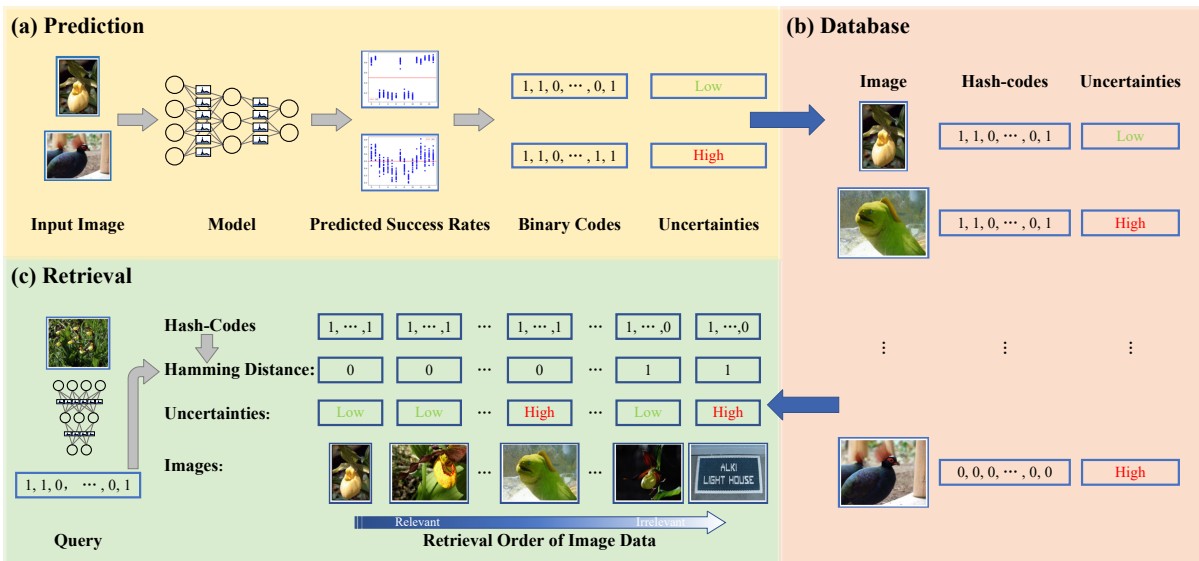

Figure 1: Schematic illustration of HashUQ with binarized uncertainties ("Low Uncertainty" v.s. "High Uncertainty"): (a) ProbHash can predict a probability distribution of Bernoulli success rates of hash-codes, from which the binary codes and the associated uncertainty can be jointly inferred. (b) Database stores both hash-codes and associated uncertainties. (c) For images with the same Hamming distance based on the derived hash-codes to the query, the retrieval order is determined by the corresponding predicted uncertainty.

Despite the tremendous amount of research conducted on representation learning, few of them aim at quantifying the uncertainty of derived representations. As DNNs have long been criticized for their tendency to make overconfident predictions and vulnerability to adversarial attacks, uncertainty quantification (UQ) in DNNs has received increasing attention and growing interest in the machine learning research community (Kendall & Gal, 2017; Abdar et al., 2021). Scalable UQ via various approximate inference methods has been developed for Bayesian Neural Networks (BNNs) (Lampinen & Vehtari, 2001; Titterington, 2004; Neal, 2012). Although they have greatly reduced the computational complexity of Markov chain Monte Carlo (MCMC) (Neal et al., 2011; Welling & Teh, 2011), few of them have been shown with promising UQ capability on representation and L2H models. Moreover, there still lacks a practical way to utilize the quantified uncertainty for improving the accuracy for downstream information retrieval.

In this paper, we propose a scalable image hashing method with UQ for image retrieval. Our main contributions can be summarized as follows:

- We develop a supervised image hashing method with uncertainty quantification capability by jointly modeling the hash-codes and neural network weights as random variables, which we term as Probabilistic Hashing (ProbHash). To the best of our knowledge, this is the first supervised image hashing algorithm with appropriately quantified uncertainty.

- To jointly model different types of uncertainty, we propose a $t$-test based measure for UQ, which provides a statistically interpretable notion of uncertainty for hashing. Our UQ method can be readily applied to various types of existing L2H models for accurate data retrieval, which we exemplify with two representative image hashing models.

- With our quantified uncertainty, we further design an image retrieval algorithm by prioritizing images with condifent hash-codes, which we term as Hashing with Uncertainty Quantification (HashUQ). Experiments on several benchmark datasets show that this consistently improves the quality of the predicted image ranking without scarifying the speed and storage consumption.

We structure our paper by first describing our probabilistic framework, ProbHash, for hashing with two representative image hashing models given in Section 2. The information supplementary to our model construction is included in *Appendix* A. We detail our $t$-test based measure of uncertainty and uncertainty aware retrieval algorithm, HashUQ, in Sections 3 and 4. Experimental evaluation by comparing with other state-of-the-art (SOTA) models and ablation studies of our methods are then presented in Section 5, with the

detailed experimental setups and supplementary results provided in *Appendix* B. Lastly, we provide literature review on Learning to Hash (L2H), the existing Uncertainty Quantification (UQ) schemes for deep learning, as well as the information bottleneck methods in Section 6. We also include a comparison between our proposed $t$-test based measure of uncertainty with Shannon's entropy in *Appendix* C, and provide retrieval examples of our developed methods in *Appendix* D. Figure 1 provides a schematic summary of HashUQ.

## 2 Probabilistic Modeling for Image Hashing

### 2.1 Mathematical Notations

Unless explicitly specified, we use the normal font letter $x$, bold letter $\boldsymbol{x}$ and bold letter in upper cases $\boldsymbol{X}$ to denote scalar, vector, and matrix, respectively. We use letter $\boldsymbol{x}$ with subscript $i, k$: $\boldsymbol{x}_{i,k}$, to represent the $k^{th}$ entry of vector $\boldsymbol{x}_i$, which is the $i^{th}$ data point in the dataset. We use the superscripts $\boldsymbol{x}^p, \boldsymbol{x}^q$ to denote a pool data point and a query, respectively. $N$, $N_{Samples}$, and $K$ denote the number of data points, number of samples, and bit-length of hash-codes, respectively.

### 2.2 Problem Settings

Suppose we have an image dataset $\{\boldsymbol{x}_1, \boldsymbol{x}_2, \ldots, \boldsymbol{x}_N\}$ with corresponding labels $\{y_1, y_2, \ldots, y_N\}$, our objective is to find a mapping $f : \mathbb{R}^{c \times h \times w} \to \{-1, 1\}^K$, which can encode all the training images $\{\boldsymbol{x}_i\}_{i=1}^N$ as well as possibly unseen query or pool images $\{\hat{\boldsymbol{x}}_j\}_{j=1}^{N'}$ into binary hash-codes $\{\boldsymbol{b}_i\}_{i=1}^N$ and $\{\hat{\boldsymbol{b}}_j\}_{j=1}^{N'}$, such that images with similar content can be encoded into hash-codes with smaller Hamming distances, and vise versa.

Recent image hashing methods (Zhu et al., 2016; Cao et al., 2017; Su et al., 2018; Yuan et al., 2020; Fan et al., 2020; Tian Hoe et al., 2021) usually adopt a feed-forward neural network with parameters $\boldsymbol{\psi}$ and binary quantization function sgn($\cdot$), which we denote as $\boldsymbol{f_\psi}$, to model the hash encoding function $\boldsymbol{f}$. Based on the learning objective and the availability of image labels, we categorize the existing image hashing methods into "Label-Target" (Cao et al., 2017; Su et al., 2018) and "Center-Target" methods (Yuan et al., 2020; Fan et al., 2020; Tian Hoe et al., 2021).

In "Label-Target" methods, the label $y_i$ for each image $\boldsymbol{x}_i$ is directly used as the learning target of binary hash-code representation $\boldsymbol{b}_i$ such that $y_i$ can be reconstructed from the binary representation with a decoder network. The "Center-Target" methods, on the other hand, minimize the distance of hash-code $\boldsymbol{b}_i$ to its "center" $\boldsymbol{c}_i$. To generate the center for each image $\boldsymbol{x}_i$, a Hadamard matrix with dimension $K$, for which we briefly review the basics in the context of hashing in *Appendix* A.1, is first constructed and each row of the matrix will be assigned to one of the $M$ classes (Yuan et al., 2020). The "center" $\boldsymbol{c}_i$ for image $\boldsymbol{x}_i$ is then the row vector corresponding to image label $y_i$. Some other randomized algorithms have also been used for the cases when $M > K$ (Yuan et al., 2020).

### 2.3 ProbHash

Learning the hashing function $\boldsymbol{f_\psi}$ can inevitably introduce different types of uncertainties to different phases in model construction, training, as well as prediction. To give an example, applying the over-parameterized DNN models will indispensably lead to the uncertainty in model parameters. Moreover, the learning targets to model are also noisy, which can be shown in a "Center-Target" setup where the centers $\{\boldsymbol{c}_i\}_{i=1}^M$ are generated either from a Hadamard matrix or some randomized algorithm and further randomly assigned to each class. This motivates us to build a probabilistic (in contrast to existing deterministic) hashing model which can consider the uncertainty with different sources and characters. We therefore develop BNN-based probabilistic hashing (ProbHash) by modeling $\boldsymbol{\psi}$ as random variables. The hash-codes $\boldsymbol{b}$ are further modeled as another Bernoulli random vector given a realization of $\boldsymbol{\psi}$ and an input image data sample $\boldsymbol{x}$. We illustrate our ProbHash modeling using probabilistic graphical model in Figure 2.

While we focus on developing our method based on two representative supervised image hashing models, ProbHash is a general framework that can be seamlessly extended to the majority of L2H models with various network structures, data types, likelihood assumptions, and various supervised and unsupervised formulations.

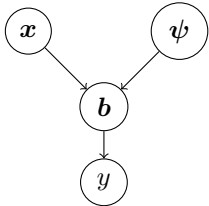

Figure 2: Illustrations of our probabilistic hashing (ProbHash) in graphical model.

Different variational families, such as dropout (Gal & Ghahramani, 2016) with a learnable or data-dependent rates (Gal et al., 2017; Boluki et al., 2020; Fan et al., 2021) and factorized Gaussian (Blundell et al., 2015), may also be adopt on $\boldsymbol{\psi}$ to achieve better flexibility. The probability distribution $p(\boldsymbol{\psi})$ and $p(\boldsymbol{b}|\boldsymbol{\psi}, \boldsymbol{x})$ can be interpreted to represent *epistemic uncertainty* and *aleatoric uncertainty* in hash-code prediction, similar as the categorization in literature (Kendall & Gal, 2017; Hüllermeier & Waegeman, 2021). Our probabilistic model for L2H can offer a more concise formulation compared to many existing methods (Zhu et al., 2016; Cao et al., 2017; Su et al., 2018; Yuan et al., 2020; Fan et al., 2020; Tian Hoe et al., 2021), which adopt different regularization tricks to minimize the quantization error in model training. Moreover, the introduction of $p(\boldsymbol{b}|\boldsymbol{\psi}, \boldsymbol{x})$ will lead to a closed-form log-likelihood function in "Center-Target" construction, which can be easily optimized without the straight-through heuristic (Bengio et al., 2013) as detailed in Section 2.4.

## 2.4 Bayesian Learning for Image Hashing

Our main objective is to learn $p(\boldsymbol{b}|\boldsymbol{x}, \boldsymbol{\psi})$, which is the conditional distribution of $\boldsymbol{b}$ by a neural network hashing function parametrized via $\boldsymbol{\psi}$ after observing image $\boldsymbol{x}$, and $p\left(\boldsymbol{\psi}|\{\boldsymbol{x}_i, y_i\}_{i=1}^N\right)$, which is the posterior distribution of neural network parameters after observing the training data $\{\boldsymbol{x}_i, y_i\}_{i=1}^N$. To jointly learn $p(\boldsymbol{b}|\boldsymbol{x}, \boldsymbol{\psi})$ and $p\left(\boldsymbol{\psi}|\{\boldsymbol{x}_i, y_i\}_{i=1}^N\right)$ in the Bayesian paradigm while avoiding the intractability of computing the marginal distribution $p(y|\boldsymbol{x})$, we resort to the amortized variational inference with the variational distributions $q(\boldsymbol{b}|\boldsymbol{x}, \boldsymbol{\psi})$ and $q(\boldsymbol{\psi})$. We learn the variational parameters $\boldsymbol{\psi}$ by maximizing the Evidence Lower BOund (ELBO):

$$
\begin{aligned}
&\log p\left(\{y_i\}_{i=1}^N | \{\boldsymbol{x}_i\}_{i=1}^N\right) \\
&\geq \mathbb{E}_{q(\boldsymbol{\psi})}\left[\sum_{i=1}^N \mathbb{E}_{q(\boldsymbol{b}_i|\boldsymbol{\psi}, \boldsymbol{x}_i)}\left[\log p\left(y_i|\boldsymbol{b}_i\right)\right] - \sum_{i=1}^N D_{KL}\left(q\left(\boldsymbol{b}_i|\boldsymbol{\psi}, \boldsymbol{x}_i\right) \| p\left(\boldsymbol{b}_i|\boldsymbol{x}_i\right)\right)\right] - D_{KL}\left(q\left(\boldsymbol{\psi}\right) \| p\left(\boldsymbol{\psi}|\{\boldsymbol{x}_i\}_{i=1}^N\right)\right).
\end{aligned} \tag{1}
$$

We provide a step-by-step derivation of Expression (1) in *Appendix* A.

We set the prior distribution $p(\boldsymbol{b}|\boldsymbol{x})$ in (1) to be a factorized Bernoulli distribution with the success probability $\boldsymbol{p}$ being a hyperparameter $p(\boldsymbol{b}|\boldsymbol{x}) = p(\boldsymbol{b}) \sim \text{Bernoulli}(\boldsymbol{b}; \boldsymbol{p})$ independent of $\boldsymbol{x}$. This will lead to the closed-form KL-divergence term in (1):

$$
D_{KL}\left(q\left(\boldsymbol{b}_i|\boldsymbol{x}_i, \boldsymbol{\psi}\right) \| p\left(\boldsymbol{b}_i\right)\right) = \sum_{k=1}^K \sigma\left(\boldsymbol{f}_{\boldsymbol{\psi}, k}\left(\boldsymbol{x}_i\right)\right) \log \frac{\sigma\left(\boldsymbol{f}_{\boldsymbol{\psi}, k}\left(\boldsymbol{x}_i\right)\right)}{\boldsymbol{p}_k} + \sigma\left(-\boldsymbol{f}_{\boldsymbol{\psi}, k}\left(\boldsymbol{x}_i\right)\right) \log \frac{\sigma\left(-\boldsymbol{f}_{\boldsymbol{\psi}, k}\left(\boldsymbol{x}_i\right)\right)}{1 - \boldsymbol{p}_k}.
$$

Next, we give our detailed model constructions and provide the specific optimization objectives for "Label-Target" and "Center-Target", respectively.

**ProbHash with Label-Target Construction:** For "Label-Target" construction, we assume the likelihood $p(y_i|\boldsymbol{b}_i)$ to be a categorical distribution parameterized by a neural network $\boldsymbol{g}_{\boldsymbol{\theta}}(\cdot)$ with the softmax activation:

$$
y \sim \text{Categorical}\left(y; \text{Softmax}\left(\boldsymbol{g}_{\boldsymbol{\theta}}\left(\boldsymbol{b}\right)\right)\right),
$$

where $\boldsymbol{\theta}$ are the parameters of the likelihood model. The variational posterior $q(\boldsymbol{b}|\boldsymbol{x}, \boldsymbol{\psi})$ is then constructed to be a factorized Bernoulli distribution parameterized by a neural network with parameters $\boldsymbol{\psi}$: $\boldsymbol{f}_{\boldsymbol{\psi}}(\cdot)$ and the sigmoid activation function $\boldsymbol{\sigma}(\cdot)$:

$$
q\left(\boldsymbol{b}|\boldsymbol{x}, \boldsymbol{\psi}\right) \sim \text{Bernoulli}\left(\boldsymbol{b}; \boldsymbol{\sigma}\left(\boldsymbol{f}_{\boldsymbol{\psi}}\left(\boldsymbol{x}\right)\right)\right).
$$

Let $\boldsymbol{W}_l$ be the neural network weights at layer $l$, and $\boldsymbol{\psi} = \{\boldsymbol{W}_l\}_{l=1}^L$. We assume the independence of the weights of each layer $q(\boldsymbol{\psi}) = \prod_{l=1}^L q(\boldsymbol{W}_l)$. For simplicity, we adopt the dropout variational distribution with the dropout rate at layer $l$ to be $p_l \in [0, 1]$, following Gal & Ghahramani (2016). The neural network weights $\boldsymbol{W}_l$ for dropout variational distribution can be written as:

$$\boldsymbol{W}_l = \boldsymbol{M}_l \circ \boldsymbol{Z}_l,$$

where $\circ$ denotes the Hadamard (entry-wise) product and $\boldsymbol{Z}_l$ is the dropout mask, each row containing either all 1s or 0s with probability $p_l$ and $1 - p_l$. $\{\boldsymbol{M}_l\}_{l=1}^L$ denote the variational parameters, which will be learned from data jointly with $\boldsymbol{\theta}$. We omit the subscript for $\{\boldsymbol{M}_l\}_{l=1}^L$ and only use $q(\boldsymbol{\psi})$ to denote the network weight variational distribution throughout our discussion. The KL-divergence of the variational posterior with the Gaussian prior $D_{KL}(q(\boldsymbol{\psi})\|p(\boldsymbol{\psi}|\{\boldsymbol{x}_i\}_{i=1}^N))$ can be approximated using a weight decay term (Gal & Ghahramani, 2016), which we also omit in the following discussion for simplicity. We also note here that there are other possible amortized variational inference solutions. As an ablation study, an empirical comparison with another widely used variational distribution family of fully factorized Gaussian weights is included in *Appendix* B.5.

With the training data $\{\boldsymbol{x}_i, y_i\}_{n=1}^N$, we solve the following loss minimization problem with the estimated ELBO:

$$\mathcal{L}_{\mathrm{ELBO}}\left(\{\boldsymbol{M}_l\}_{l=1}^L, \boldsymbol{\theta}\right) = \mathbb{E}_{q(\boldsymbol{\psi})}\left[\sum_{i=1}^N -\mathbb{E}_{q(\boldsymbol{b}_i|\boldsymbol{x}_i, \boldsymbol{\psi})}\left[\log p_{\boldsymbol{\theta}}\left(y_i|\boldsymbol{b_i}\right)\right] + \lambda \sum_{i=1}^N D_{KL}\left(q\left(\boldsymbol{b}_i|\boldsymbol{x}_i, \boldsymbol{\psi}\right)\|p\left(\boldsymbol{b}_i|\boldsymbol{x}_i\right)\right)\right], \quad (2)$$

where $\lambda$ is a hyperparameter reflecting the trade-off between model fitness to the training data and discrepancy to the prior distribution.

**ProbHash with Center-Target Construction:** To derive ProbHash for existing "Center-Target" methods (Yuan et al., 2020), we assume the likelihood distribution $p(y_i|\boldsymbol{b}_i)$ proportional to the exponential of the Hamming distance between $\boldsymbol{b}_i$ and the predefined "center" vector for $i^{th}$ data, $\boldsymbol{c}_i$:

$$p\left(y_i|\boldsymbol{b}_i\right) = p\left(\boldsymbol{c}_i|\boldsymbol{b}_i\right) = \frac{\exp\left(-\phi \mathcal{D}_{\mathrm{Hamming}}\left(\boldsymbol{c}_i, \boldsymbol{b}_i\right)\right)}{Z}, \quad (3)$$

where $Z$ is the normalizing constant. A discussion and comparison with other likelihood choices is included in *Appendix* A.3. The negative ELBO minimization objective $\mathcal{L}'_{\mathrm{ELBO}}(\boldsymbol{\psi})$ can be similarly derived to be:

$$\mathcal{L}'_{\mathrm{ELBO}}\left(\boldsymbol{\psi}\right) = \mathbb{E}_{q(\boldsymbol{\psi})}\left[\sum_{i=1}^N -\mathbb{E}_{q(\boldsymbol{b}_i|\boldsymbol{x}_i, \boldsymbol{\psi})}\left[\log p\left(\boldsymbol{c}_i|\boldsymbol{b}_i\right)\right] + \lambda \sum_{i=1}^N D_{KL}\left(q\left(\boldsymbol{b}_i|\boldsymbol{x}_i, \boldsymbol{\psi}\right)\|p\left(\boldsymbol{b}_i|\boldsymbol{x}_i\right)\right)\right]. \quad (4)$$

Notice that by further incorporating (3) into (4), the log-likelihood term in the negative ELBO has the following closed form:

$$\begin{aligned}
&\mathbb{E}_{q(\boldsymbol{b}_i|\boldsymbol{x}_i, \boldsymbol{\psi})}\left[-\log p(\boldsymbol{c_i}|\boldsymbol{b_i})\right] \\
=&\mathbb{E}_{q(\boldsymbol{b}_i|\boldsymbol{x}_i, \boldsymbol{\psi})}\left[\phi|\boldsymbol{c}_i - \boldsymbol{b}_i| - \log Z\right] \\
=&\mathbb{E}_{q(\boldsymbol{b}_i|\boldsymbol{x}_i, \boldsymbol{\psi})}\left[\phi \sum_k^K |c_{i,k} - b_{i,k}| - \log Z\right] \\
=&\phi \sum_k^K \left[-2c_{i,k}\boldsymbol{\sigma}\left(\boldsymbol{f}_{\boldsymbol{\psi},k}\left(\boldsymbol{x}_i\right)\right) + c_{i,k} + 1\right] - \log Z,
\end{aligned}$$

where $\log Z$ is a constant irrelevant to the variational parameters $\boldsymbol{\psi}$. This closed-form log-likelihood can be easily estimated and optimized with any package equipped with automatic differentiation. Our negative ELBO minimization objective with both log-likelihood and KL-divergence terms in closed-forms can lead to more principled training with better retrieval accuracy, compared to existing works (Zhu et al., 2016; Cao et al., 2017; Su et al., 2018; Yuan et al., 2020; Fan et al., 2020; Tian Hoe et al., 2021) where model optimization

by biased gradient estimation of the discontinuous function $\frac{\partial \boldsymbol{b}}{\partial \boldsymbol{\psi}}$ with the heuristic straight-through (ST) trick (Bengio et al., 2013) can lead to degraded performance, as shown empirically in Section 5.4.1.

**Connection to Information Bottleneck:** While originally derived from the perspective of Bayesian inference, our proposed ProbHash with "Label-Target" construction can be considered as a realization of variational information bottleneck (Alemi et al., 2016) on Learning-to-Hash with a Bernoulli latent representation and a Bayesian neural network encoder. Minimizing the loss (2), which has a similar format as the variational information bottleneck objective in Alemi et al. (2016), is also equivalent to maximizing the mutual information between latent hash-codes $\boldsymbol{b}$ and labels $y$ while minimizing the mutual information between latent hash-codes $\boldsymbol{b}$ and inputs $\boldsymbol{x}$.

## 2.5 Predicting the Binary Hash-codes

By solving the optimization problems discussed in Section 2.4, we obtain two variational posteriors $q_{\boldsymbol{\psi}}(\boldsymbol{b}|\boldsymbol{x}, \boldsymbol{\psi})$ and $q(\boldsymbol{\psi})$. Our goal is to find a mapping from input data to a binary vector for image hashing. We calculate the probability distribution of hash-code $\hat{\boldsymbol{b}}$ given new test image $\hat{\boldsymbol{x}}$ by marginalizing out the model parameters $\boldsymbol{\psi}$:

$$q\left(\hat{\boldsymbol{b}}|\hat{\boldsymbol{x}}\right) = \int q\left(\hat{\boldsymbol{b}}|\hat{\boldsymbol{x}}, \boldsymbol{\psi}\right) q\left(\boldsymbol{\psi}\right) d\boldsymbol{\psi}, \tag{5}$$

where $q\left(\hat{\boldsymbol{b}}|\hat{\boldsymbol{x}}\right)$ is a Bernoulli distribution whose success probability can be estimated using Monte-Carlo sampling. With $N_{Sample}$ samples of neural network parameters $\{\boldsymbol{\psi}_n\}_{n=1}^{N_{Sample}}$, the Bernoulli success probability of hash-code can also be sampled: $\{\boldsymbol{\sigma}\left(\boldsymbol{f}_{\boldsymbol{\psi}_n}\left(\hat{\boldsymbol{x}}\right)\right)\}_{n=1}^{N_{Sample}}$, and the marginal distribution of hash-code distribution can be estimated by:

$$q\left(\hat{\boldsymbol{b}}|\hat{\boldsymbol{x}}\right) = \text{Bernoulli}\left(\hat{\boldsymbol{b}}; \mathbb{E}_{\boldsymbol{\psi}}\left[\boldsymbol{\sigma}\left(\boldsymbol{f}_{\boldsymbol{\psi}}\left(\hat{\boldsymbol{x}}\right)\right)\right]\right) \approx \text{Bernoulli}\left(\hat{\boldsymbol{b}}; \frac{\sum_{n=1}^{N_{Sample}} \boldsymbol{\sigma}\left(\boldsymbol{f}_{\boldsymbol{\psi}_n}\left(\hat{\boldsymbol{x}}\right)\right)}{N_{Sample}}\right).$$

We use the *maximum-a-posterior* (MAP) estimation of the variational posterior $q(\hat{\boldsymbol{b}}|\hat{\boldsymbol{x}})$ to predict the most probable hash-code of the input image. This is equivalent to thresholding the predicted Bernoulli success probability at 0.5:

$$\underset{\hat{\boldsymbol{b}} \in \{-1,1\}^K}{\text{argmax}} \ q\left(\hat{\boldsymbol{b}}|\hat{\boldsymbol{x}}\right) = \mathbf{1}_{\mathbb{E}_{\boldsymbol{\psi}}[\boldsymbol{\sigma}(\boldsymbol{f}_{\boldsymbol{\psi}}(\hat{\boldsymbol{x}}))] \geq 0.5} - \mathbf{1}_{\mathbb{E}_{\boldsymbol{\psi}}[\boldsymbol{\sigma}(\boldsymbol{f}_{\boldsymbol{\psi}}(\hat{\boldsymbol{x}}))] < 0.5}.$$

# 3 Quantifying Uncertainties in Hashing

## 3.1 Uncertainties in Hashing

Our probabilistic modeling for both $\boldsymbol{\psi}$ and $\boldsymbol{b}|\boldsymbol{x}, \boldsymbol{\psi}$ leads to a hierarchically constructed distribution for hash-code $\boldsymbol{b}$ with two groups of uncertainties, as mentioned in Section 2.3. In particular, a low uncertainty on $\boldsymbol{\psi}$ in our learned model, with $q(\boldsymbol{\psi})$ being similar to a Dirac delta function, indicates high confidence over model parameters. On the other hand, given $\boldsymbol{\psi}$ and $\boldsymbol{x}$, the predicted Bernoulli success probability closer to 0 and 1 or 0.5 respectively means the model is certain or uncertain of the specific entry of hash-codes. Suppose we have $N_{Sample}$ samples of neural network parameters $\{\boldsymbol{\psi}_n\}_{n=1}^{N_{Sample}}$ with each $\boldsymbol{\psi}_n \sim q(\boldsymbol{\psi})$. For the ease of discussion, we will use $\pi_{n,k}(\hat{\boldsymbol{x}})$ to denote the Bernoulli success rate of $k^{th}$ entry given input image $\hat{\boldsymbol{x}}$ with $n^{th}$ sample $\boldsymbol{\psi}_n$:

$$\pi_{n,k}(\hat{\boldsymbol{x}}) = \boldsymbol{\sigma}(\boldsymbol{f}_{\boldsymbol{\psi}_n,k}(\hat{\boldsymbol{x}})).$$

Consider four different cases when the mean value of $\pi_{\cdot,k}$ to be either 0.6 or 0.9, with either low or high variance. If a model consistently predicts $\pi_{\cdot,k} \approx 0.9$, this indicates the model clearly towards predicting $\hat{\boldsymbol{b}}_{\cdot,k}$ to be "1". In either cases when predicting 0.6 consistently or 0.9 on average with high variance, the model is still in favor of deciding on "1" but with more uncertainty. In the last case when the model outputs with the mean value of $\pi_{\cdot,k}$ to be 0.6 with high variance, the model is less confidence for the decision on "1". This motivates us to quantify the uncertainty by conducting hypothesis testing on the samples of predicted Bernoulli success rates of hash-codes, which we detail in Section 3.2.

## 3.2 Measure of Uncertainty by $t$-test

For any of the $k^{th}$ entry of a hash-code $b_{\cdot,k}$, the significant difference between the Bernoulli success and failure probabilities indicates confident prediction while having similar success and failure probabilities implies uncertain prediction. Given the sampled probabilities of $b_{\cdot,k}$ being "1": $\{\pi_{n,k}\}_{n=1}^{N_{Sample}}$ and "0": $\{1 - \pi_{n,k}\}_{n=1}^{N_{Sample}}$, we conduct *student's t-test* of the null hypothesis that the mean probabilities of being "1" and "0" are equal on each of the $k^{th}$ entry. We represent the uncertainty in $b^k$ using the $p$-value of the test, which we denote as $P_k(\hat{\boldsymbol{x}})$. A smaller $p$-value in favor of rejecting the null hypothesis indicates a more significant difference between the Bernoulli success and failure rates, reflecting our confidence in $b_{\cdot,k}$. The total uncertainty on the hash-code given an input image $\hat{\boldsymbol{x}}$ is represented using the summed log $p$-value over all the $K$ entries: $\sum_{k=1}^{K} \log P_k(\hat{\boldsymbol{x}})$. The summed log $p$-value over all entries provides an aggregated measure of hash-code uncertainty for one specific input data $\hat{\boldsymbol{x}}$, which can also be interpreted as the logarithm of the joint tail probability given the null hypothesis under the factorization assumption.

We adopt the *paired sample t-test* considering the dependency of the success and failure rates. This test is also equivalent to performing *one sample t-test* of the null hypothesis that the sample mean of Bernoulli success rate equals 0.5, which can be interpreted similarly.

## 3.3 Discussion

The idea of using statistic hypothesis testing as the measure of uncertainty is not totally new, and has already been adopted in some previous works in other applications (Fan et al., 2021), which we find to be potentially suitable for quantifying the uncertainties of hash-codes when the two aforementioned uncertainties are presented together. The above metric also matches our intuition about uncertainties in L2H discussed in Section 3.1, as the $t$-statistics becomes higher when the Bernoulli success rates $\boldsymbol{\pi}^k$ is less variant and closer to 1 or 0, which will further lead to the lower tail probabilities and $p$-values, and implies a higher confidence. In *Appendix* C, we provide detailed analysis and numerical comparison between our proposed paired sample $t$-test based measure with Shannon's entropy based measure of uncertainty. In *Appendix* E, we further provide histograms and Q-Q plots of the difference between paired samples of each Bernoulli success rate of hash-code vectors generated using 5 query images from the ImageNet dataset, which validate the assumption for paired sample $t$-test that the difference between the paired values is normally distributed.

# 4 Hashing with Uncertainty Quantification

## 4.1 Hashing in the Presence of Uncertainty

Consider a database of $N_p$ data entries $\{\boldsymbol{x}_1^p, \boldsymbol{x}_2^p, \ldots, \boldsymbol{x}_{N_p}^p\}$, with $N_p$ on the order of millions or even trillions with the corresponding hash-codes $\{\boldsymbol{b}_1^p, \boldsymbol{b}_2^p, \ldots, \boldsymbol{b}_{N_p}^p\}$. When performing nearest neighbor search for the query $\boldsymbol{x}^q$ with code $\boldsymbol{b}^q$, compared to the data entries which are predicted to be relevant to the query but with high uncertainty, we prefer entries which are relevant based on the derived hash-codes with high confidence. This motivates us to take the quantified uncertainties into account for retrieval. We propose our uncertainty aware retrieval to prioritize the data entries with confident hash-codes while deprioritizing the data samples with uncertain hash-codes, which we term as Hashing with Uncertainty Quantification (HashUQ).

In particular, when the query $\boldsymbol{x}^q$ is provided along with the hash-code $\boldsymbol{b}^q$, we rank each entry $\boldsymbol{x}_i^p$ in the database based on the Hamming distance between $\boldsymbol{b}^q$ and $\boldsymbol{b}_i^p$, which we denote as $D_{\text{Hamming}}(\boldsymbol{b}^q, \boldsymbol{b}_i^p)$, from low to high. In the meanwhile, we rank each entry based on quantified uncertainty associated with hash-code. The retrieval is performed by first comparing the Hamming distance $D_{\text{Hamming}}(\boldsymbol{b}^q, \boldsymbol{b}_i^p)$. When we encounter two entries with the same Hamming distance to the query, we further compare the quantified uncertainties of hash-codes and the entry with more confident hash-code will be retrieved with higher priority. We can equivalently describe our method as retrieving a pool data sample $\boldsymbol{x}_i^p$ in a sequential order based on the sorting results of the following expression in the ascending order over all the pool data in the database:

$$D_{\text{Hamming}}(\boldsymbol{b}^q, \boldsymbol{b}_i^p) + \alpha \sum_{k=1}^{K} \log P_k(\boldsymbol{x}_i^p),$$

where $\alpha$ is a positive real number small enough such that $|\alpha \sum_{k=1}^{K} \log P_k(\hat{x})| < 1$. Notice that $\alpha$ here is not a hyper-parameter in either our model or algorithm. Given the fact that the length of hash-code $K$ is typically much smaller than $N_p$, thousands of database entries whose hash-codes have exactly the same Hamming distance to the query when performing nearest neighbor search can now be retrieved with the model confidence also taken into account.

### 4.2 Efficient Storage of Uncertainties

Our proposed test on Bernoulli samples measures uncertainty in a real value, which is typically stored in the "floating-point" format in computer systems. Considering that the L2H is typically applied to the retrieval system which is sensitive to the storage space and running time, the space for storing the quantified uncertainty is unignorable compared to the hash-code which takes only tens or hundreds of bits.

We address this practical challenge by discretizing the quantified uncertainties. A simple example is to quantize the uncertainties into binary, with each hash-code to be either "Low Uncertainty" or "High Uncertainty" namely. When we discretize the quantified uncertainties into $d$ quantized levels, the storage space reduces to $\log_2 d$-bit complexity.

### 4.3 Computational Complexity

Assume that we have a total of $N$ data samples in a database and $M$ queries for retrieval. For each query, only $r$ data samples need to be retrieved from the database and ranked accordingly, with $r \ll N$. When the vanilla Hamming-distance-based hashing strategy is applied, the total computational complexity for $M$ retrievals is $\mathcal{O}(C_1 M N)$ for Hamming distance computation and $\mathcal{O}(C_2 M N \log N)$ for sorting, with $C_1, C_2$ being two constants and $C_1 \gg C_2$. As the quantified uncertainty is only associated with each data point and does not need to be computed each time when the retrieval is performed, our uncertainty-based ranking strategy will introduce almost no additional computation overhead to sort the quantified uncertainty when multiple retrieved images have the same distance to the query.

### 4.4 Discussion

**Why Prioritizing Confident Pool Data:**  Given a confident pool data sample and an uncertain data sample with the predicted Hash-codes having the same Hamming distance to the query. The uncertain pool data is more likely to be either more relevant to the query, or irrelevant to the query compared to the confident one. However, given the fact that these Hamming distances are small when we are performing nearest neighbor search, it is more likely that the latter one hold true, and we should choose the confident pool data sample to achieve better retrieval accuracy, which we validate experimentally in *Appendix* B.6.

**Introduced Computation Consumption:**  Different from other applications of machine learning where their inference time will be the main concern regarding computational budget, the hash-codes of pool data can be pre-computed and stored in a hashing-based retrieval setup, and therefore the retrieval time and storage consumption are the main computational concerns. The inference time of the query data will remain the same as other non-probabilistic methods as we are not quantifying the uncertainty of query data. While we mainly focus on using the uncertainty values of pool data throughout the discussion above, the uncertainty values of query data can also be used to help improving retrieval accuracy either in a scenario where the computational resources allow, or by adopting some other uncertainty measures which does not require intensive sampling, such as Shannon's Entropy. We show in Section 5.3 that the query data with higher hash-code uncertainty will be more likely to retrieve irrelevant documents, which can be potentially combined with the aforementioned method to warn the users of the potentially erroneous retrievals.

## 5 Experiments

We empirically evaluate our ProbHash and HashUQ, comparing with other SOTA L2H methods. We first describe the experimental setup for performance evaluation by providing the evaluation matrices and baseline models in Sections 5.1 and 5.2. We empirically evaluate our ProbHash and compare with other SOTA L2H

Table 1: Comparison of HashUQ with both SOTA "Label-Target" and "Center-Target" baselines on ImageNet, MS COCO, and NUS WIDE datasets. The best performing models for each setup (Except for full HashUQ) are illustrated in bold font. * denotes the overall best performing model.

| Target | Method | ImageNet(mAP@1000) | | | MS Coco(mAP@5000) | | | NUS WIDE@5000 | | |
|---|---|---|---|---|---|---|---|---|---|---|
| | | 16 bits | 32 bits | 64 bits | 16 bits | 32 bits | 64 bits | 16 bits | 32 bits | 64 bits |
| Label | GreedyHash | 0.611 | **0.657** | **0.680** | 0.654 | 0.702 | 0.714 | 0.776 | 0.796 | 0.812 |
| | HashNet | 0.335 | 0.502 | 0.607 | 0.612 | 0.661 | 0.714 | 0.752 | **0.808** | *0.844 |
| | ProbHash | 0.595 | 0.645 | 0.673 | 0.653 | 0.703 | 0.726 | 0.775 | 0.800 | 0.816 |
| | HashUQ-Bin | **0.618** | 0.653 | 0.675 | **0.663** | **0.709** | **0.729** | **0.790** | 0.806 | 0.818 |
| | HashUQ | 0.623 | 0.655 | 0.678 | 0.669 | 0.712 | 0.730 | 0.799 | 0.812 | 0.822 |
| Center | CSQ | 0.595 | 0.668 | 0.699 | 0.651 | 0.732 | 0.769 | 0.789 | 0.820 | 0.834 |
| | OthorHash | 0.587 | 0.669 | 0.712 | 0.624 | 0.689 | 0.718 | 0.792 | 0.825 | **0.841** |
| | HSWD | 0.593 | 0.675 | 0.700 | 0.662 | 0.736 | *0.771 | 0.788 | 0.824 | 0.835 |
| | ProbHash | 0.606 | 0.680 | 0.706 | 0.665 | 0.731 | 0.764 | 0.790 | 0.820 | 0.835 |
| | HashUQ-Bin | *0.623 | *0.690 | *0.714 | *0.681 | *0.739 | 0.768 | *0.805 | *0.829 | 0.839 |
| | HashUQ | 0.631 | 0.696 | 0.718 | 0.688 | 0.743 | 0.771 | 0.814 | 0.834 | 0.843 |

methods as well as the UQ capability by HashUQ in Section 5.3. Lastly in Section 5.4.1, we analyze the effects of different model components and hyper-parameter sensitivity. Additionally in *Appendix* B, we provide a detailed description of benchmark datasets and model training, as well as the supplementary results on model evaluation and uncertainty quantification.

## 5.1 Evaluation Metrics

**Mean Average Precision:** In ranking-based retrieval, Average Precision (AP) is the integrated performance measure jointly considering precision and recall. Let $P(r)$ be the precision of the top $r$ images, and rel$(r)$ the indicator function that the $r$-th image is relevant, and $|\{\text{Relevant}\}|$ the number of all the relevant images to the query in the database. We have:

$$AP@r = \frac{\sum_{r=1}^{N} P(r) \times \text{rel}(r)}{|\{\text{Relevant}\}|}.$$

We use the *mean Average Precision@r* (mAP@$r$) on the whole test set to evaluate the retrieval performance of our uncertainty aware image hashing, with $r$ retrieved images when we calculate the average precision. Following previous works (Cao et al., 2017; Su et al., 2018; Yuan et al., 2020; Fan et al., 2020; Tian Hoe et al., 2021), we set $r = 1000$ for ImageNet, and $r = 5000$ for both MS COCO and NUS WIDE.

## 5.2 Baseline Models

For each of the "Label-Target" and "Center-Target" method, we implement our ProbHash model based on "GreedyHash" (Su et al., 2018) and "CSQ" (Yuan et al., 2020), respectively. We test ProbHash with and without HashUQ, and denote each of them as "ProbHash" and "HashUQ" throughout this section. The retrieval algorithm with model confidence by HashUQ has been discussed in Section 4. As HashUQ takes significantly more storage space compared to all other methods, we further binarize the uncertainties quantified using the paired sample $t$-test as detailed in Section 4.2, which we denote as "HashUQ-Bin". We compare with the following SOTA baselines: HashNet (Cao et al., 2017), GreedyHash (Su et al., 2018), CSQ (Yuan et al., 2020), OrthoHash (Tian Hoe et al., 2021), and HSWD (Doan et al., 2022). We reproduce the results of GreedyHash, CSQ, and HSWD, and implement HashUQ performance evaluation and comparison based on the implementation from DeepHash-pytorch[1] with PyTorch 1.8.1 (Paszke et al., 2019). We reproduce OrthoHash results using the implementation from its official GitHub repository[2] (Tian Hoe et al., 2021).

---

[1] https://github.com/swuxyj
[2] https://github.com/kamwoh/orthohash

### 5.3 Experimental Evaluation of Proposed Methods

**ProbHash Achieves Competitive Performances:** Table 1 summarizes the empirical performance of baseline models and different implementations of our proposed model. Compared to the corresponding baseline models, "GreedyHash" and "CSQ", our ProbHash achieves competitive retrieval accuracy. Across two constructions and three datasets, our ProbHash achieves similar or better performance on five setups and only perform slightly worse in some of the experiments under one setup (Label-Target on ImageNet).

**ProbHash Enables Uncertainty Quantification:** To show that our ProbHash can provide reasonable uncertainty estimation for the predicted hash-code of a supervised image hashing model, we provide the boxplot of retrieval accuracy for the query images with respect to the quantile of their predicted hash-code uncertainty with both "Center-Target" and "Label-Target" construction with hash-code length $K = 32$ on ImageNet in Figure 3. The corresponding plots for $K = 16$ and $K = 64$ are included in Figure 5 of Appendix B.2. In bothFigures 3a and 3b, we can clearly observe that the predicted uncertainties are typically small for hash-codes of query image with high retrieval accuracy. The median, 25% and 75% quantiles of retrieval accuracy will all decrease as the predicted hash-code uncertainties of query images increase, which shows that a high estimated uncertainty will be typically associated with an erroneous prediction.

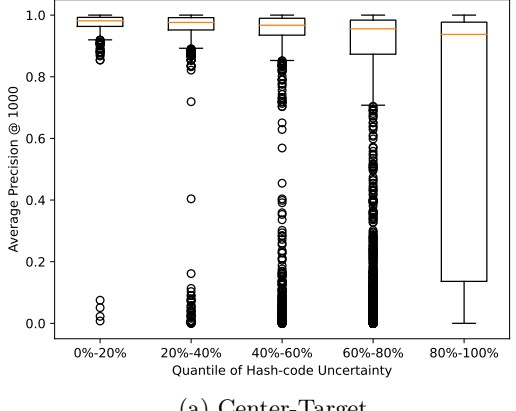
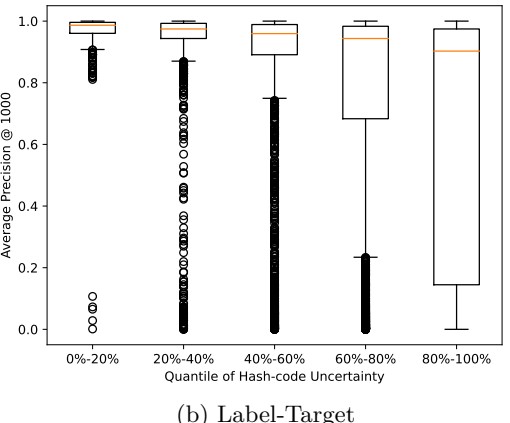

(a) Center-Target  (b) Label-Target

Figure 3: Boxplot of retrieval accuracy with respect to the quantile of hash-code uncertainty with $K = 32$ of our ProbHash with "Center-Target" and "Label-Target" constructions on ImageNet.

**HashUQ Further Achieves State-of-the-art Retrieval Accuracy:** By prioritizing data with high confidence, our HashUQ-Bin consistently delivers superior performance across nearly all tested settings of the 'Label-Target' and 'Center-Target' constructs, demonstrating the broad applicability of our method. HashUQ-Bin under "Center-Target" can achieve prominent retrieval performance with the best retrieval accuracy in 7 of the 9 settings tested, and is also comparable to the best performing models in the other two settings. Our HashUQ-Bin improves the retrieval accuracy of the corresponding ProbHash model with a similar amount as the full HashUQ, demonstrating its practical effectiveness exploiting the benefits from UQ.

**Discussion:** HashUQ is especially effective for retrievals with short hash-codes, as there will typically be more retrieved data samples whose hash-codes have the same Hamming distance from the query, compared to retrievals with long hash-codes. Although we have only shown the results with two representative L2H baseline models, our ProbHash and HashUQ can be combined with other advanced neural network architectures (Yuan et al., 2020), improved likelihood function design (Tian Hoe et al., 2021), and more sophisticated regularization strategies (Tian Hoe et al., 2021; Doan et al., 2022). We expect similar empirical performance improvement. In Section 5.4.4, we demonstrate that HashUQ outperforms baseline models in retrieval results, even when these models utilize more bits than HashUQ. This further validates the superior performance of HashUQ-Bin relative to the baselines.

Table 2: Retrieval accuracy of our ProbHash with KL-regularization and HashUQ included and excluded, trained with different gradient estimators on ImageNet. We set $\lambda = 0$ for experiments without a checkmarker on $\lambda$ and $\lambda = 1.0$ otherwise.

| Component | | | ImageNet(mAP@1000) | | |
|---|---|---|---|---|---|
| Gradient Estimator | $\lambda$ | HashUQ | 16 bits | 32 bits | 64 bits |
| Straight-through | | | 0.573 | 0.660 | 0.689 |
| Straight-through | ✓ | | 0.592 | 0.666 | 0.693 |
| Closed-form | ✓ | | 0.604 | 0.679 | 0.707 |
| Closed-form | ✓ | ✓ | **0.631** | **0.696** | **0.718** |

## 5.4 Ablation Studies

### 5.4.1 Effects of Different Model Components

We study the effects of each component of our model and different optimization strategies by running the experiments with different components included and excluded. All the ablation experiments are performed on "Center-Target" implementation. The corresponding retrieval performances on ImageNet are reported in Table 2. We use "Straight-through" to denote a model trained with the gradient of $\frac{\partial \boldsymbol{b}}{\partial \boldsymbol{\psi}}$ estimated by the Straight-Through trick, and "Closed-form" to denote the same model trained by optimizing the closed-form ELBO in Section 2.4. Each component—closed-form ELBO optimization, prior regularization, and UQ-based ranking strategy—contributes to improved retrieval accuracy. Optimal retrieval performance is achieved when all these elements are combined.

### 5.4.2 Sensitivity of Hyperparameter $\lambda$

Our model only introduces one more hyperparameter $\lambda$ for the trade-off between data belief and prior. To study the sensitivity of our model performance with respect to $\lambda$, we run our model with different $\lambda$ values on ImageNet, with the corresponding retrieval accuracy reported in Figure 4a. As long as $\lambda$ is set to be within an appropriate range, the retrieval accuracies of ProbHash and HashUQ are not sensitive to $\lambda$ . Moreover, our UQ based ranking strategy can consistently improve the retrieval accuracy with all different $\lambda$ values and all different hash-code bit-length $K$'s.

### 5.4.3 Comparison of Different UQ Measures

To compare different UQ measures in terms of the effectiveness in improving retrieval accuracy, we evaluate the UQ-based retrieval strategy using different UQ measures. The results on different datasets with $K$-bit hash-codes are reported in Table 3. More information about other two measures and a comparison can be found in Appendix B.4. While the uncertainty quantified in all of the three measures can help improve the retrieval accuracy, our $t$-test-based UQ consistently performs the best among all the measures on all three datasets with different $K$'s. A possible explanation is the full consideration of two types of uncertainties in the adopted $p$-value compared to other measures. For example, Shannon's Entropy mostly just considers the uncertainty of $\boldsymbol{b}$, while the summed variance only considers the uncertainty of $\boldsymbol{\psi}$. This further justifies our approach for the L2H task and our preference for the $t$-test-based UQ methodology.

### 5.4.4 Effects of Number of Quantization Levels

We further study the effects of the number of quantization levels $d$ discussed in Section 4.2 on the retrieval accuracy with our UQ-based ranking strategy. We determine the quantization levels by making the same number of data samples to be discretized to each of the quantization levels. Figure 4b plots the retrieval accuracy by ProbHash and HashUQ with different numbers of quantization levels when $K = 32$. We also include the performance of ProbHash with 2 more bits for each case to compare the performance within the same storage. The corresponding figures with $K = 16$ and $K = 64$ can be found in *Appendix* B.3. While storing the quantified uncertainty with a higher precision can lead to more accurate retrievals, with the best

Table 3: Retrieval accuracy of our UQ-based retrieval with different UQ measures: $t$-test based metric (denoted as "$t$-test"), Shannon's Entropy ("Entropy"), and the summed variance of Bernoulli success probability ("Variance") on ImageNet, MS COCO and NUS WIDE datasets.

| Dataset | UQ Measure | 16 bits | 32 bits | 64 bits |
|---------|-----------|---------|---------|---------|
| ImageNet | Entropy | 0.628 | 0.695 | 0.717 |
| | Variance | 0.617 | 0.688 | 0.712 |
| | $t$-test | **0.631** | **0.696** | **0.718** |
| MS Coco | Entropy | 0.682 | 0.740 | 0.769 |
| | Variance | 0.673 | 0.735 | 0.766 |
| | $t$-test | **0.688** | **0.743** | **0.771** |
| NUS WIDE | Entropy | **0.814** | 0.833 | 0.842 |
| | Variance | 0.795 | 0.824 | 0.837 |
| | $t$-test | **0.814** | **0.834** | **0.843** |

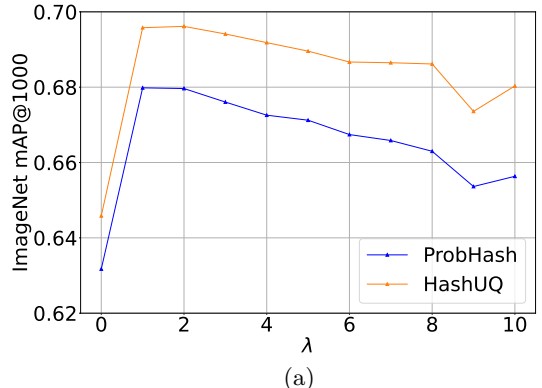

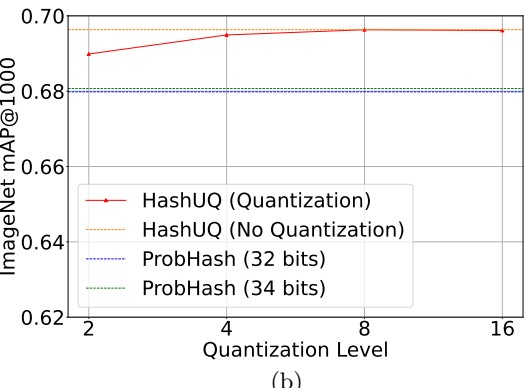

Figure 4: Retrieval accuracy of HashUQ with (a) different coefficient $\lambda$ values (b) quantization levels with $K = 32$ on ImageNet.

performance when no quantization is implemented, our UQ-based ranking method can bring significant and consistent performance improvement to the implementation only considering the distances. They achieve better performance even with the binarized or quaternarized uncertainty, which only takes 1 or 2 extra bits for storage and almost no extra computational overhead.

When we train a deep hashing model with a few more bits but use it in the same way without considering the quantified uncertainty, the benefit brought by the extra bits is almost marginal compared to uncertainty aware retrieval, as can be clearly observed in Figure 4b. As no model re-training is needed for HashUQ as long as the base model being a probabilistic model, our HashUQ can be applied to systems whose storage allows us to have additional memory to include more quantization levels, which adds on the flexibility as more spareable resources can be utilized to improve the accuracy and user experience. The reason that the extra bits may bring almost no benefit to data retrieval can be possibly explained as the retrieval accuracy of an L2H model can be highly dependent on the minimum pairwise Hamming distance between pre-specified "center" vectors (Tian Hoe et al., 2021) over all pairs, whose largest possible value will rarely change when $K$ only increases for a few more bits[3].

---

[3]https://www.win.tue.nl/ aeb/codes/binary-1.html

# 6   Related Work

**Information Retrieval and Learning to Hash:**   Given query data $\{\boldsymbol{x}_1^q, \boldsymbol{x}_2^q, \dots, \boldsymbol{x}_M^q\}$ and a database with $N$ data points $\{\boldsymbol{x}_1^p, \boldsymbol{x}_2^p, \dots, \boldsymbol{x}_N^p\}$, the main objective of information retrieval is to generate a ranking of all the pool data $\boldsymbol{\rho}_i^q(\boldsymbol{x}_i^q; \{\boldsymbol{x}_j^p\}_{j=1}^N)$ for all the query data point, such that pool data similar to the query can get higher ranking order while pool data different from the query will be ranked lower in each of the predicted ranking. The Learning-to-Hash (L2H) based information retrieval methods achieve this by learning an encoding function $\boldsymbol{f}$ which can map the query and all of the pool data into $K$-bit binary hash-codes $\{\boldsymbol{b}_1^p, \boldsymbol{b}_2^p, \dots, \boldsymbol{b}_N^p\}$ (Weiss et al., 2008; Salakhutdinov & Hinton, 2009; Strecha et al., 2011; Liu et al., 2011; 2014; Li et al., 2016). The retrieval is then performed by ranking all the pool data points according to the Hamming distance of their hash codes to the hash code of the query.

Pioneering works of L2H using traditional handcraft features include Spectral Hashing (Weiss et al., 2008), Linear Discriminant Analysis (LDA) Hashing (Strecha et al., 2011), and Graph Hashing (Liu et al., 2011; 2014). Semantic Hashing (Salakhutdinov & Hinton, 2009) is among the earliest works using DNNs for hashing, where a two-stage procedure is developed to train a deep auto-encoder for document retrieval in a fully unsupervised manner. For high-dimensional complex data such as images and video, some early deep hashing models include Deep Hashing (DH) (Erin Liong et al., 2015), Deep Pairwise-Supervised Hashing (DPSH) (Li et al., 2016), and Self-Supervised Temporal Hashing (SSTH) (Zhang et al., 2016). Recent research on image hashing mostly focuses on solving the discrete optimization problem for principled training (Cao et al., 2017; Su et al., 2018), reducing the information loss in code quantization (Doan et al., 2022), improving the loss function for orthogonal and disentangled hash code generation (Tian Hoe et al., 2021; Doan et al., 2022), and designing novel and flexible learning schemes (Kang et al., 2019; Yuan et al., 2020; Fan et al., 2020).

**Uncertainty Quantification in Deep Learning:**   The primary notion of uncertainty in deep learning is to use a probability distribution as the model prediction instead of a point estimation. Most of uncertainty can be categorized into two major groups based on the sources and characteristics (Kendall & Gal, 2017; Hüllermeier & Waegeman, 2021): *aleatoric uncertainty*, which represents the uncertainty due to the intrinsic randomness of the physical process and is typically modeled by either the softmax outputs in classification tasks or the Gaussian distributed outputs in regression tasks (Nix & Weigend, 1994); and *epistemic uncertainty*, which is the uncertainty due to the lack of knowledge and is typically modeled by replacing the frequentist DNNs with Bayesian Neural Networks (BNNs) (Lampinen & Vehtari, 2001; Titterington, 2004; Neal, 2012). BNNs model network parameters or activations as random variables whose distributions are learned through the posterior updates using Bayes' theorem. To solve the intractability of posterior inference in many situations, various approximate inference methods based on Markov Chain Monte Carlo (MCMC) (Neal et al., 2011; Welling & Teh, 2011) and variational inference (VI) (Blei et al., 2017) have been developed, including the Monte Carlo dropout (MC Dropout) (Gal & Ghahramani, 2016; Gal et al., 2017; Boluki et al., 2020; Fan et al., 2021) and Bayes-By-Backprop (Blundell et al., 2015). Some UQ methods developed from a frequentist's point of view include conformal prediction (Angelopoulos & Bates, 2021) and quantile regression (Koenker, 2005), with the point estimation replaced by an interval representing the confidence of the model prediction.

Our work differs from previous works of learning-to-hash in a way that our work is the first probabilistic framework for supervised hashing. We also propose a new measure of uncertainty designed for hashing based on student's $t$-test which can simultaneously quantify the aleatoric and epistemic uncertainties. Additionally, our work is also one of the few works that demonstrate the potential applicability of quantified uncertainties on improving real-world applications.

# 7   Conclusion and Discussion

In this paper, we propose ProbHash, a probabilistic framework for modeling hashing function in image retrieval along with a $t$-test based measure of uncertainty which jointly takes different sources of uncertainty into account. We further develop HashUQ, an uncertainty aware hashing strategy, and show that considering the quantified uncertainty is an effective and efficient way to enhance retrieval accuracy with little computation and storage overhead.

One drawback of our current work lies in the fact that we approximate the posterior of network parameters and hash-code using the variational distributions with the mean-field assumption to make it tractable, which has been shown to possibly underestimate the variance (Blei et al., 2017). Although adopting a deep neural network with its advanced representational capacity and flexibility can mitigate this issue, future research directions include extending to more complex variational distributions with fewer restrictions, and exploring diffusion models' capabilities in uncertainty quantification (Han et al., 2022) for hashing.

**Broader Impact Statement**

This paper presents a novel approach for image retrieval, contributing to the ongoing advancements in the intersection of machine learning, computer vision, and information retrieval. Our primary goal is to providing the capability of uncertainty quantification to hashing models, and further enhancing the efficiency and accuracy of retrieval systems. By doing so, we aim for valuable tools in applications ranging from content organization to visual search.

The potential broader impact of our work extends to various societal dimensions. Improved image retrieval systems can have positive impact for fields such as content management, recommendation, and searching systems. However, we also recognize the responsibility associated with deploying such technology, especially concerning privacy, bias, and ethical considerations related to image content.

**Acknowledgments**

This work was supported in part by the U.S. National Science Foundation (NSF) grants SHF-2215573, IIS-2212418, and IIS-2212419. Portions of this research were conducted with the advanced computing resources provided by Texas A&M High Performance Research Computing.

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

In the appendix, we first include a brief introduction to Hadamard Matrices, which are commonly used to construct hashing centers in previous "Center-Target" methods in Section A.1. We then provide the step-by-step derivation of the training loss function for ProbHash with "Label-Target" construction and experimental comparison with other likelihood choices in Section A.2 and Section A.3, respectively. We also include detailed evaluation pipeline, additional experimental results along with an empirical comparison of different variational distributions for Bayesian neural networks in Section B. In Section D we provide comparison of some exemplar image retrieval results with and without uncertainty quantification. Lastly in Section E, we provide histograms and Q-Q plots of the difference between paired samples of each Bernoulli success rate of hash-code vectors generated using 5 query images from the ImageNet dataset to support the applicability of paired samples $t$-test on image hashing problems.

## A  SUPPLEMENTARY INFORMATION FOR MODEL CONSTRUCTIONS

### A.1  A Brief Introduction to Hadamard Matrices and Its Application on Learning-to-Hash

A Hadamard matrix is a square matrix of dimension $2^k, k \in \mathbb{Z}^+$ which can be defined by induction:

$$
\begin{aligned}
H_1 &= \begin{bmatrix} 1 \end{bmatrix}, \\
H_2 &= \begin{bmatrix} 1 & 1 \\ 1 & -1 \end{bmatrix}, \\
H_{2^k} &= \begin{bmatrix} H_{2^{k-1}} & H_{2^{k-1}} \\ H_{2^{k-1}} & -H_{2^{k-1}} \end{bmatrix}.
\end{aligned}
\tag{6}
$$

Each row or column of a Hadamard matrix is orthogonal to all other rows or columns. The row vectors of matrix $\begin{bmatrix} H \\ -H \end{bmatrix}$, where $H$ is a Hadamard matrix of dimension $2^k$, will form a set of $2^k$-dimensional vectors with the theoretically maximum minimum mutual Hamming distance, which has been empirically shown to affect the retrieval accuracy of "Center-Target" methods (Tian Hoe et al., 2021). These Hadamard matrices have been used to construct the hashing centers in previous "Center-Target" methods (Yuan et al., 2020; Tian Hoe et al., 2021; Doan et al., 2022).

### A.2  Derivation of Expression (1) in the Main Text

We provide the detailed derivation of Expression (1) in Section 2.4 of the *Main Text*. The derivations corresponding to the "Label-Target" construction and "Center-Target' construction can be easily obtained by replacing $p(\{y_i\}_{i=1}^N | \{\boldsymbol{x}_i\}_{i=1}^N, \boldsymbol{\psi})$ with $p_{\boldsymbol{\theta}}(\{y_i\}_{i=1}^N | \{\boldsymbol{x}_i\}_{i=1}^N, \boldsymbol{\psi})$ and $p(\{\boldsymbol{c}\}_{i=1}^N | \{\boldsymbol{x}_i\}_{i=1}^N, \boldsymbol{\psi})$.

$$
\begin{aligned}
&\log p(\{y_i\}_{i=1}^N | \{\boldsymbol{x}_i\}_{i=1}^N) \\
\geq& \mathbb{E}_{q(\boldsymbol{\psi})} \left[ \log \int p(\{y_i\}_{i=1}^N | \{\boldsymbol{x}_i\}_{i=1}^N, \boldsymbol{\psi}) \right] - D_{KL} \left( q(\boldsymbol{\psi}) || p(\boldsymbol{\psi} | \{\boldsymbol{x}_i\}_{i=1}^N) \right) \\
=& \mathbb{E}_{q(\boldsymbol{\psi})} \left[ \log \int p(\{y_i\}_{i=1}^N, \{\boldsymbol{b}_i\}_{i=1}^N | \{\boldsymbol{x}_i\}_{i=1}^N, \boldsymbol{\psi}) d\{\boldsymbol{b}_i\}_{i=1}^N \right] - D_{KL} \left( q(\boldsymbol{\psi}) || p(\boldsymbol{\psi} | \{\boldsymbol{x}_i\}_{i=1}^N) \right) \\
=& \mathbb{E}_{q(\boldsymbol{\psi})} \left[ \log \int p(\{y_i\}_{i=1}^N | \{\boldsymbol{b}_i\}_{i=1}^N) p(\{\boldsymbol{b}_i\}_{i=1}^N | \{\boldsymbol{x}_i\}_{i=1}^N, \boldsymbol{\psi}) d\{\boldsymbol{b}_i\}_{i=1}^N \right] - D_{KL} \left( q(\boldsymbol{\psi}) || p(\boldsymbol{\psi} | \{\boldsymbol{x}_i\}_{i=1}^N) \right) \\
\geq& \mathbb{E}_{q(\boldsymbol{\psi})} \left[ \sum_{i=1}^N \mathbb{E}_{q(\boldsymbol{b}_i | \boldsymbol{\psi}, \boldsymbol{x}_i)} \left[ \log p(y_i | \boldsymbol{b}_i) \right] - \sum_{i=1}^N D_{KL}(q(\boldsymbol{b}_i | \boldsymbol{\psi}, \boldsymbol{x}_i) || p(\boldsymbol{b}_i | \boldsymbol{x}_i)) \right] - D_{KL} \left( q(\boldsymbol{\psi}) || p(\boldsymbol{\psi} | \{\boldsymbol{x}_i\}_{i=1}^N) \right).
\end{aligned}
$$

### A.3  Choice of Likelihood for "Center-Target" Construction

In Bayesian statistics, the likelihood represents our beliefs about what data we expect to see for each setting of parameters of the model. The choice of likelihood distribution will depend on the data type as well as the

convenience of computation. The likelihood function should take the higher value when data is consistent with the model while the lower value when the data is inconsistent with the model. For example, the isotropic Gaussian distribution is typically chosen for continuous data, and the categorical distribution is typically used for modeling the discrete data. Both belong to the exponential family and have easy-to-compute log-likelihood function. We choose the likelihood distribution:

$$p(\boldsymbol{c}_i|\boldsymbol{b}_i) \propto \exp(-\phi D_{Hamming}(\boldsymbol{c}_i, \boldsymbol{b}_i))$$

for the following reasons:

- **This probability distribution matches with our aforementioned principle for the likelihood choice:** Both $\boldsymbol{c}_i$ and $\boldsymbol{b}_i$ are binary vectors and the Hamming distance is indeed a proper choice of distance measure for binary vectors. The adopted likelihood distribution reflects our belief that the hash-code center $\boldsymbol{c}_i$ corresponding to its class-label will be more likely to have small Hamming distance to the unobserved hash-code $\boldsymbol{b}_i$. Our adopted likelihood distribution takes the highest value when $\boldsymbol{c}_i$ matches with $\boldsymbol{b}_i$ and decreases when $\boldsymbol{c}_i$ gradually deviates from $\boldsymbol{b}_i$.

- **This probability distribution belongs to the commonly adopted likelihood distributions:** Just like Gaussian or categorical distribution, our adopted likelihood distribution is a Boltzmann distribution, which is one of the most widely used probability distribution in statistical analysis and machine learning modeling. Our adopted distribution also belongs to the exponential family, which has a simple form of log likelihood function.

- **This probability distribution has computational advantages over other choices of likelihood distributions:** This specific choice of our likelihood distribution will also lead to the closed-form optimization objective as emphasized in Section 2.4 of our manuscript. The benefit of this closed-form objective is two-fold: (1) it reduces the computational consumption for conducting multiple forward Monte Carlo samples; (2) it is unbiased and will reduce the variance for stochastic gradient descent, which will lead to more efficient model training.

We compare our adopted likelihood distribution with another Boltzmann distribution:

$$p(c_i|b_i) \propto \exp(-\phi D_{Hamming}^2(c_i, b_i))$$

This distribution takes a similar format as Gaussian distribution, and the closed-form log-likelihood can also be derived as:

$$
\begin{aligned}
&\mathbb{E}_{q(\boldsymbol{b}_i|\boldsymbol{x}_i,\boldsymbol{\psi})} \left[ -\log p(\boldsymbol{c_i}|\boldsymbol{b_i}) \right] \\
=&\mathbb{E}_{q(\boldsymbol{b}_i|\boldsymbol{x}_i,\boldsymbol{\psi})} \left[ \phi|\boldsymbol{c}_i - \boldsymbol{b}_i|^2 - \log Z \right] \\
=&\mathbb{E}_{q(\boldsymbol{b}_i|\boldsymbol{x}_i,\boldsymbol{\psi})} \left[ \phi\sum_{k}^{K} |c_{i,k} - b_{i,k}|^2 - \log Z \right] \\
=&\phi\sum_{k}^{K} \left[ c_{i,k}^2 + 2c_{i,k} + 1 - 4c_{i,k}\boldsymbol{\sigma} \left( \boldsymbol{f}_{\boldsymbol{\psi},k}\left(\boldsymbol{x}_i\right)\right)^2 \right] - \log Z.
\end{aligned}
\tag{7}
$$

This means that this distribution also satisfies all the three principles for selecting likelihood function mentioned previously. The main difference between the distribution above and the likelihood distribution we adopt in the main paper is how the probability decays as the Hamming distance gets larger. We run supplementary experiments on ImageNet dataset to compare the distribution above ($D_{Hamming}^2$) and the likelihood distribution we adopt in the main paper($D_{Hamming}$), and include the results in Table 4: We observe the retrieval accuracy to be slightly affected with the likelihood distribution in the main paper replaced with the likelihood discussed above. The experiment results also show that our HashUQ strategy can help improving the retrieval accuracy regardless of the likelihood choice.

| Method | 16 bit | 32 bit | 64 bit |
|---|---|---|---|
| ProbHash-$D_{Hamming}$ | 0.606 | 0.680 | 0.706 |
| HashUQ-$D_{Hamming}$ | 0.631 | 0.696 | 0.718 |
| ProbHash-$D^2_{Hamming}$ | 0.606 | 0.678 | 0.705 |
| HashUQ-$D^2_{Hamming}$ | 0.631 | 0.692 | 0.716 |

Table 4: Comparison of different likelihood distributions for "Center-Target" Construction.

# B  SUPPLEMENTARY INFORMATION FOR EXPERIMENTS

## B.1  Supplementary Information for Evaluation Pipeline

### B.1.1  Datasets

We empirically evaluate image retrieval performances based on different supervised hashing methods to demonstrate our uncertainty aware HashUQ's superiority on three benchmark image datasets: **ImageNet** (Deng et al., 2009), **MS COCO** (Lin et al., 2014), and **NUS WIDE** (Chua et al., July 8-10, 2009). Each dataset has been further split into training, query and pool (database) sets, with the details given below:

- **ImageNet** is a large-scale image dataset (Deng et al., 2009), which contains more than 10,000,000 images labeled with 20,000 different synsets in WordNet. We follow Cao et al. (2017) and use the images from 100 selected categories for training as well as evaluation.
- **MS COCO** is an image dataset with more than $200,000$ labeled images from 80 categories (Lin et al., 2014). Multiple labels are typically associated with one image instance, which makes the retrieval task typically more challenging than ImageNet. We follow Zhu et al. (2016) and use a total of 132,218 data samples for training and evaluation.
- **NUS WIDE** is an image dataset with a total of $269,648$ data samples collected from Flickr, labeled with 81 ground-truth concepts (Chua et al., July 8-10, 2009). We follow Cao et al. (2017) and use images from the 21 most frequent concepts for model development and testing, with each concepts containing at least 5000 images.

One major difference between our evaluation pipeline and those adopted in previous works is that we explicitly include a validation set for model selection while most of the previous works does not differentiate the validation set with the test set (Cao et al., 2017; Su et al., 2018; Yuan et al., 2020; Fan et al., 2020; Tian Hoe et al., 2021). The dataset statistics with the adopted data splits are summarized in Table 5.

Table 5: Statistics of data splits in ImageNet, MS COCO, and NUS WIDE datasets.

| Dataset | Dev. | | Test | | # cls |
|---|---|---|---|---|---|
| | Train. | Val. | Query | Pool | |
| ImageNet | 11700 | 1300 | 5000 | 128,503 | 100 |
| MS Coco | 9000 | 1000 | 5000 | 117,218 | 80 |
| NUS WIDE | 9000 | 1500 | 2100 | 193,734 | 21 |

### B.1.2  Hyperparameters and Training Details

We use the AlexNet (Krizhevsky et al., 2012) backbone for all the experiments to be consistent with the evaluation pipelines of previous works Cao et al. (2017); Su et al. (2018); Yuan et al. (2020); Tian Hoe et al. (2021); Doan et al. (2022). Three extra fully connected layers with the latent dimensions 4096 are added at the end of neural network backbone. We regularize the last three layers with dropout in each of the hidden layer for all the methods benchmarked, with the neurons reweighed deterministically in baselines and dropped randomly in our ProbHash and HashUQ at test time. The the dropout rates are set to be 0.5. We optimize the neural networks using RMSprop (Hinton et al., 2012) optimizer with the learning rate $1e-5$ and weight decay $1e-5$ for all the models. We use our derived closed-form ELBO as the minimization objective

for "Center-Target" construction and estimated ELBO using Monte-Carlo sampling as the minimization objective for "Label-Target" construction with the Straight-Through (ST) trick to estimate the gradient of discontinuous function $\frac{\partial \boldsymbol{b}}{\partial \boldsymbol{\psi}}$. We choose fair Bernoulli distribution $p(\boldsymbol{b}_{\cdot,k} = 1) = p(\boldsymbol{b}_{\cdot,k} = -1) = 0.5$ as the prior for each entry of the hash-codes and set $\alpha$ as $\max_i |\sum_{k=1}^{K} \log P_k(\boldsymbol{x}_i^p)|$ in all of our simulation. The coefficient balancing between data-fitting and prior belief $\lambda$ is set to be 1.0 unless specified. Notice that the $\phi$ in "center-target" construction can be absorbed into $\lambda$, and we use a fixed $\phi = 2$ throughout the paper without loss of generality. We perform model evaluation using 100 samples from the learned dropout variational distribution. The best performing model on the validation sets during the first 100 training epochs are chosen to be evaluated on query and pool datasets.

## B.2 Supplementary Results for Uncertainty Quantification

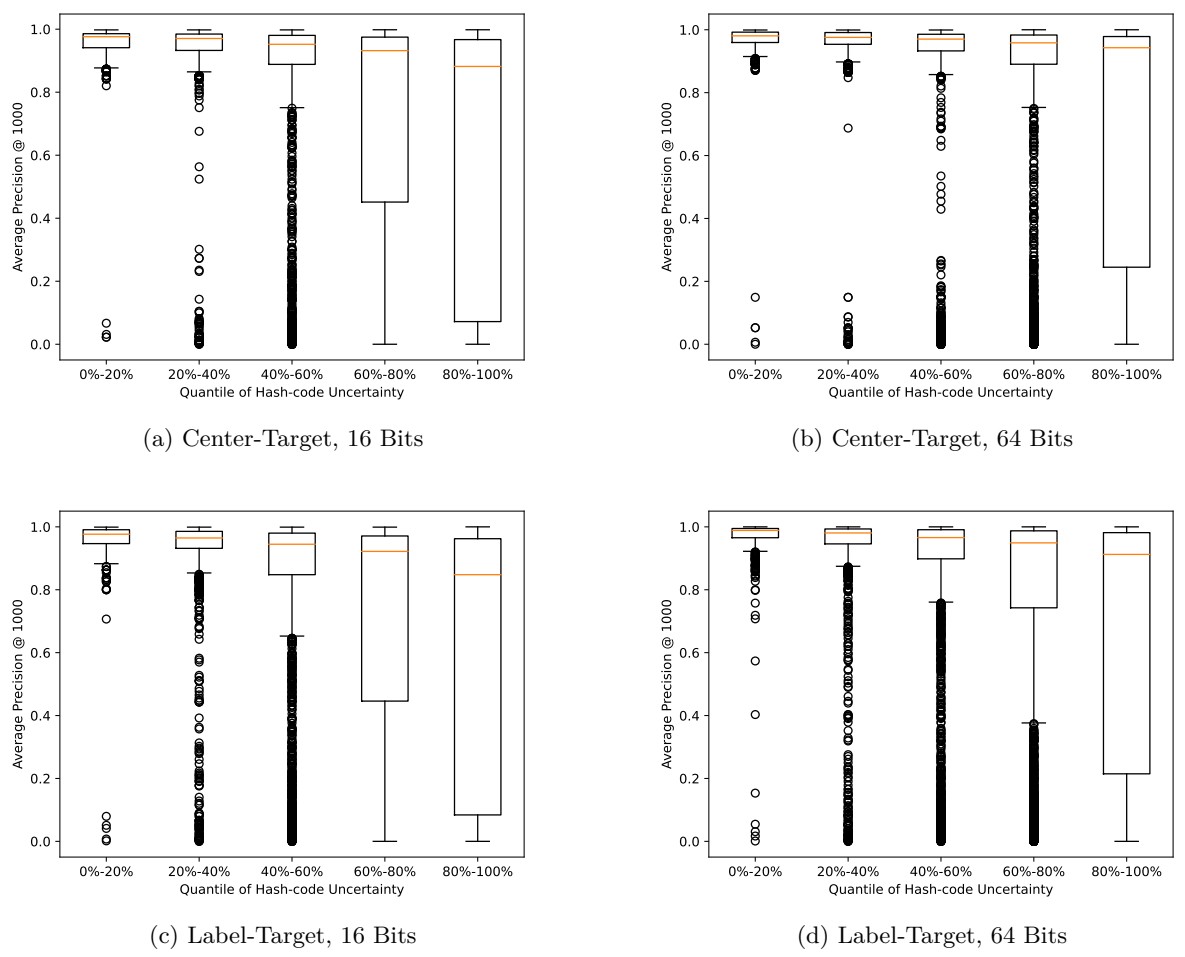

(a) Center-Target, 16 Bits

(b) Center-Target, 64 Bits

(c) Label-Target, 16 Bits

(d) Label-Target, 64 Bits

Figure 5: Boxplot of retrieval accuracy with respect to the quantile of hash-code uncertainty with $K = 16$, $K = 32$ and $K = 64$ of our ProbHash with "Center-Target" and "Label-Target" constructions on ImageNet.

We present more results showing the uncertainty quantification capability of our proposed method supplementary to experiments in Sections 5.3 of the *Main Text*. The boxplots of retrieval accuracy for the query images with respect to the quantile of their predicted hash-code uncertainty with different constructions with $K = 16$ and $K = 64$ are provided in Figure 5. We can observe similar trend as in the *Main Text*: A higher

estimated uncertainty of hash-code will be typically predicted with the present of a query with erroneous retrieval results.

## B.3 Supplementary Results for Ablation Experiments

We include more experimental results supplementary to the experiments in Sections 5.4.2 and 5.4.4 of the *Main Text*. We plot the retrieval accuracy with respect to different coefficients $\lambda$ in Figures 6a and 6b, and the ones with different numbers of quantization levels in Figures 6c and 6d on the ImageNet dataset with $K = 16$ and $K = 64$. We observe a similar trend as in Sections 5.4.2 and 5.4.4 of the *Main Text*: Both of our ProbHash and HashUQ provide retrieval performances insensitive to $\lambda$, as long as $\lambda$ is set within a appropriate range. Our HashUQ can consistently provide performance improvement over ProbHash regardless of $\lambda$, $K$ and quantization level settings. Our HashUQ also brings more retrieval accuracy improvement compared to a model without considering the quantified uncertainty as the allowed bits increase.

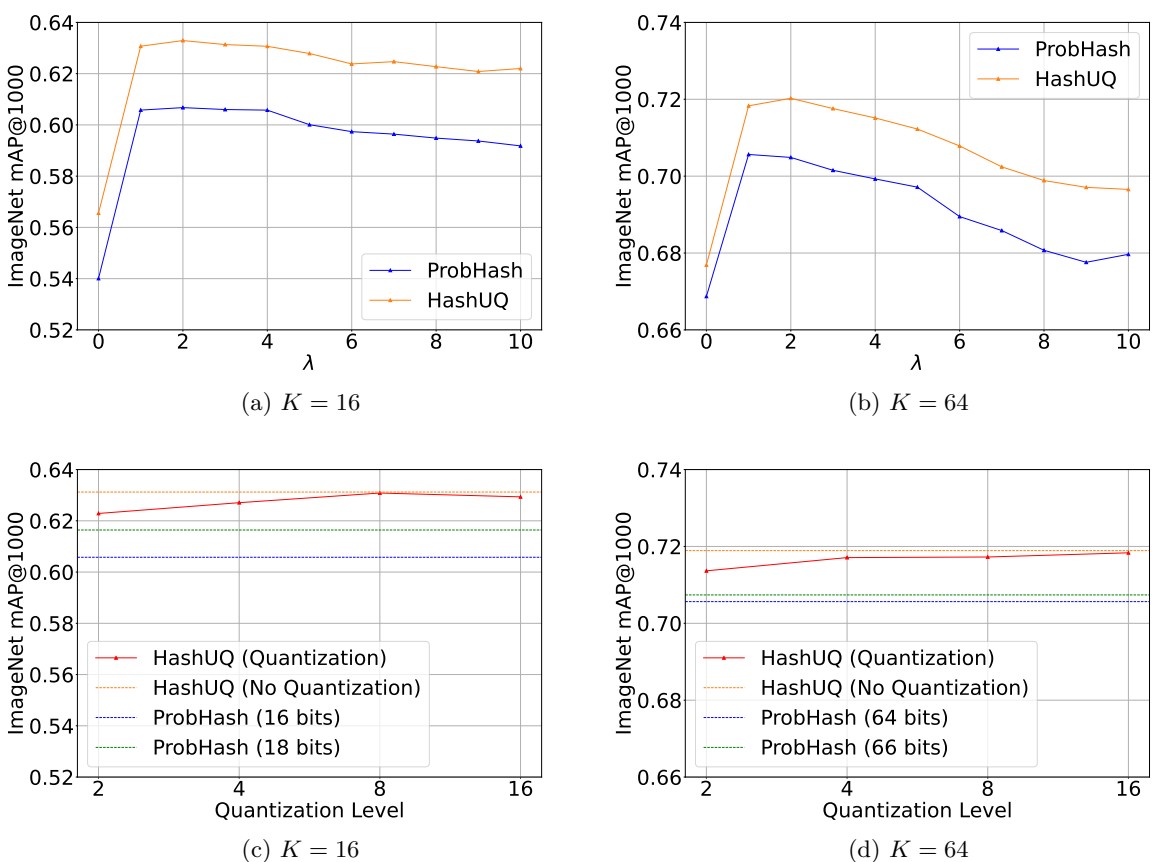

(a) $K = 16$

(b) $K = 64$

(c) $K = 16$

(d) $K = 64$

Figure 6: Retrieval accuracy with respect to different (a) coefficient $\lambda$ values with $K = 16$ (b) number of quantization levels with $K = 16$ (c) coefficient $\lambda$ values with $K = 64$ (d) number of quantization levels with $K = 64$ on ImageNet.

## B.4 Supplementary Information for Different Metrics

We here provide more information about the comparative study of different metrics for measuring the uncertainties in hashing complementary to Section 5.4.3 of the *Main Text*. Given a bunch of Bernoulli success rates $\{\boldsymbol{\pi_n^k}\}_{n=1}^{N_{Sample}}$, one way to measure the uncertainty is to calculate the variance of $\boldsymbol{\pi_n^k}$: $\mathbb{E}_{\psi}[(\boldsymbol{\pi}^k - \mathbb{E}_{\psi}[\boldsymbol{\pi}^k])^2]$. This type of uncertainty measure will mostly quantify the effect of the uncertainty of variational distribution

| Method | 16 bit | 32 bit | 64 bit |
|---|---|---|---|
| ProbHash-FFG | 0.597 | 0.647 | 0.662 |
| ProbHash-MCD | 0.606 | 0.680 | 0.706 |
| HashUQ-FFG | 0.633 | 0.677 | 0.686 |
| HashUQ-MCD | 0.631 | 0.696 | 0.718 |
| HashUQ-FFG-Bin | 0.627 | 0.673 | 0.678 |
| HashUQ-MCD-Bin | 0.623 | 0.690 | 0.714 |

Table 6: Comparison of Fully Factorized Gaussian weights (FFG) and MC Dropout (MCD) inference for ProbHash and HashUQ on ImageNet dataset.

of model parameters $\boldsymbol{\psi}$ on the Bernoulli success rates $\boldsymbol{\pi^k}$. We report the experimental results with proposed HashUQ based retrieval on different image datasets in Table 3, which we denote as "Variance".

Another way to quantify the uncertainty considering the adopted factorized Bernoulli distribution for hash-code is the Shannon's Entropy, which has been adopted in Wang et al. (2023). As no uncertainty of $\boldsymbol{\psi}$ is considered in Wang et al. (2023), here we generalize the metrics for our method and consider the Shannon's Entropy of $q(\hat{\boldsymbol{b}}|\hat{\boldsymbol{x}}, \boldsymbol{\psi})$: $H(q(\hat{\boldsymbol{b}}|\hat{\boldsymbol{x}}, \boldsymbol{\psi})) = \mathbb{E}_{q(\boldsymbol{\psi})}[\mathbb{E}_{q(\hat{\boldsymbol{b}}|\hat{\boldsymbol{x}}, \boldsymbol{\psi}))}[-\log q(\hat{\boldsymbol{b}}|\hat{\boldsymbol{x}}, \boldsymbol{\psi})]]$. This will measure the uncertainty of hash-code $\boldsymbol{b}^k$ averaged over the distribution of neural network parameters $q(\boldsymbol{\psi})$. We also test this metric with the results denoted as "Entropy" in Table 3. In both of the above cases, we sum the variance and entropy over all the $K$ entries to have an aggregated notion of uncertainty for hash-code of one input image considering the factorized assumptions of our variational distribution.

## B.5 Comparison of Different Variational Distributions for Bayesian Neural Networks

Last but not least, we provide an empirical comparison between dropout, the variational distribution we adopted, to another widely used distribution for Bayesian neural network approximation: Fully Factorized Gaussian weights (FFG). For each entry $\boldsymbol{W}_{l,i,j}$ of neural network weight matrix $\boldsymbol{W}_l$, the independent Gaussian distribution is assumed:

$$\boldsymbol{W}_{l,i,j} \sim \mathcal{N}\left(\boldsymbol{W}_{l,i,j}; \mu_{l,i,j}, \sigma^2_{l,i,j}\right),$$

with $\{\mu_{l,i,j}, \sigma_{l,i,j}\}$ as the variational parameters. We specifically adopt the Bayes-By-Backprop (Blundell et al., 2015) implementation for Fully Factorized Gaussian weights, in which the variational distribution is reparameterized using the standard Gaussian distribution and variational parameters optimized by maximizing the Evidence Lower BOund (ELBO) of training data through gradient-based optimization. We model the weight matrices of the last $\{1, 2, 3\}$ fully connected layers as factorized Gaussian and set the learning rates to be $\{1 \times 10^{-5}, 5 \times 10^{-5}, 1 \times 10^{-4}\}$. A standard Gaussian prior is adopted and $\{1, 5, 10\}$ Monte Carlo samples are used to reduce the variance of gradient estimation, with the best performing model in the aforementioned configurations on ImageNet dataset choosen to be reported in Table 6.

The quantified uncertainties of Fully Factorized Gaussian weights can similarly be used to help enhance the retrievals as the dropout weights. We observe worse empirical performance of Fully Factorized Gaussian weights compared to dropout in most of the setups with or without using uncertainties. In the meanwhile, the performance improvement due to uncertainties quantified from the Fully Factorized Gaussian weights is slightly more significant compared to dropout. While some more flexible distribution families may better approximate the posterior and model the uncertainties (Foong et al., 2020), it also brings computational challenges in optimization, which may require further analysis on the ELBO maximization objective and gradient based optimization methods.

## B.6 Comparison between Two Strategies: Prioritizing Confident Pool Data or Prioritizing Uncertain Pool Data

In the Table 7, we include the retrieval accuracy of our ProbHash, HashUQ with the confident pool data prioritized (HashUQ-Confident) and HashUQ with the uncertain pool data prioritized (HashUQ-Uncertain)

| Method | 16 bit | 32 bit | 64 bit |
|---|---|---|---|
| ProbHash | 0.606 | 0.680 | 0.706 |
| HashUQ-Confident | 0.631 | 0.696 | 0.718 |
| HashUQ-Uncertain | 0.563 | 0.656 | 0.689 |

Table 7: Comparison between two strategies: prioritizing confident pool data or prioritizing uncertain pool data.

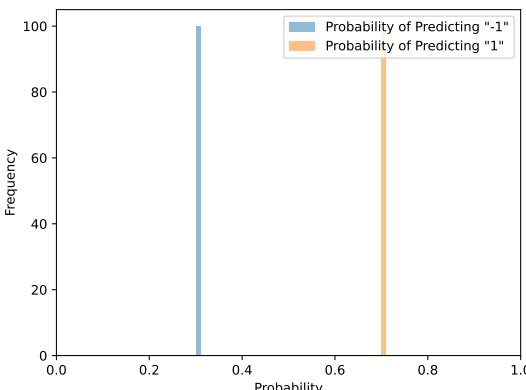 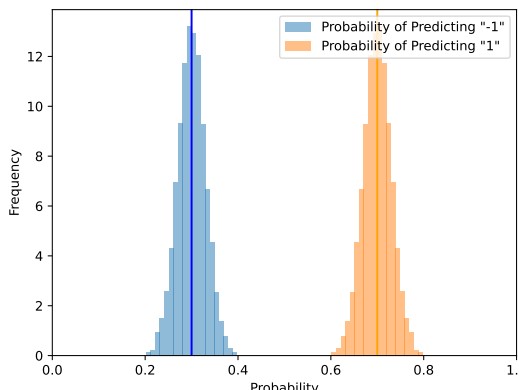

(a) Case A: the posterior samples always predict the probabilities 0.3 and 0.7.

(b) Case B: the values of posterior samples fluctuating from $(0.2, 0.8)$ to $(0.4, 0.6)$.

Figure 7: Illustration of exemplar scenario considered where the $t$-test can be effectively used to rank uncertainty.

on ImageNet. We observe worse retrieval accuracy of HashUQ-Uncertain compared to HashUQ-Confident and ProbHash. This implies that the uncertain pool data is more likely to be irrelevant to the query compared to not only the confident ones, but also most of the other pool data in between when Hamming distances are small.

## C  SUPPLEMENTARY ANALYSIS OF UNCERTAINTY MEASURES

### C.1  Comparison between $t$-test and Shannon's entropy based measures of uncertainty

The paired sample $t$-test measures whether the population mean of two groups of samples are equal. Here we connect the estimated uncertainty by $t$-test statistics to Shannon's entropy based measure of uncertainty. We present an exemplar scenario where the two-sample t-test can be effectively used to rank uncertainty:

- In Case A, for a given input, the posterior samples always predict the probabilities of predicting "-1" and "1" as $(\pi'_{\cdot,k}, \pi''_{\cdot,k}) = (0.3, 0.7)$.

- In Case B, while the average probabilities remain $(0.3, 0.7)$, the values of $(\pi'_{\cdot,k}, \pi''_{\cdot,k})$ fluctuate, ranging from $(0.2, 0.8)$ to $(0.4, 0.6)$.

Figure 7 illustrate the two cases in the scenario considered. Clearly, the uncertainty in Case B is greater, and the two-sample t-test captures this difference effectively while Shannon's entropy do not capture this increased uncertainty in Case B.

**Paired sample $t$-test based Uncertainty Measure:** We conducted the paired sample $t$-test with the null hypothesis $H_0$: $\overline{\pi'_{\cdot,k}} = \overline{\pi''_{\cdot,k}}$. The $t$-statistic of our adopted paired sample test is calculated as follows:

$$t(\pi'_{\cdot,k}, \pi''_{\cdot,k}) = \frac{\overline{\pi'_{\cdot,k} - \pi''_{\cdot,k}}}{\mathrm{Std}(\pi'_{\cdot,k} - \pi''_{\cdot,k})/\sqrt{N}},$$

where $\overline{\pi'_{\cdot,k} - \pi''_{\cdot,k}}$ and $\mathrm{Std}(\pi'_{\cdot,k} - \pi''_{\cdot,k})$ represent the sample mean and standard deviation of $(\pi'_{n,k} - \pi''_{n,k})$. $N$ is the number of samples. The uncertainty of the $k$-th entry is quantified as the tail probability of observing $t(\pi_{\cdot,k})$ under the null hypothesis $H_0$:

$$pval(t(\pi'_{\cdot,k}, \pi''_{\cdot,k})) = P(t((\tilde{\pi'}_{\cdot,k}, \tilde{\pi''}_{\cdot,k})) \geq t(\pi'_{\cdot,k}, \pi''_{\cdot,k})|(\tilde{\pi'}_{\cdot,k}, \tilde{\pi''}_{\cdot,k}) \sim H_0).$$

We use the log scale, $\log pval(t(\pi'_{\cdot,k}, \pi''_{\cdot,k}))$ specifically as the measure of uncertainty for $k$-th entry.

**Uncertainty Measured using Conditional Shannon's Entropy:** We calculated the conditional Shannon's entropy as follows:

$$\mathbb{E}_{\boldsymbol{\psi}\sim q(\boldsymbol{\psi})}[H(b|\psi, x)] \approx -\overline{\left[\pi'_{\cdot,k} \log \pi'_{\cdot,k} + (1 - \pi'_{\cdot,k}) \log(1 - \pi'_{\cdot,k})\right]}.$$

Some previous works use this conditional Shannon's entropy to measure *aleatoric uncertainty.*

**Uncertainty Measured using Total Shannon's Entropy:**

$$H(b|x) \approx -\left[\overline{\pi'}_{\cdot,k} \log \overline{\pi'}_{\cdot,k} + (1 - \overline{\pi'}_{\cdot,k}) \log(1 - \overline{\pi'}_{\cdot,k})\right].$$

Some previous works use this total Shannon's entropy to measure *total uncertainty.* An induced *epistemic uncertainty* can be calculated by substracting total Shannon's Entropy to conditional Shannon's entropy:

$$H(b|x) - \mathbb{E}_{\psi\sim q(\psi)}[H(b|\psi, x)].$$

**Connection and Difference:** Both of the $t$-test and Shannon's entropy give the highest estimation of uncertainty when the model consistently predicts $\pi_{\cdot,k} \approx 0.5$ (corresponding to high aleatoric uncertainty) while the lowest estimation of uncertainty when the model predicts $\pi_{\cdot,k} \approx 0$ or $\pi_{\cdot,k} \approx 1$ (corresponding to low aleatoric uncertainty). When the Shannon's Entropy based measure of uncertainty mentioned above is adopted, the total uncertainty do not change and the aleatoric uncertainty decrease as the variation of $\pi$ increase. The uncertainty measured using $t$-test will increase when the variation of $\pi$ increase. We include a numerical comparison between paired sample $t$-test and Shannon's entropy based measures of uncertainty using 100 samples from truncated Gaussian distributions in Figure 8 and 9 to show how each of these two uncertainties change as the mean $\overline{\pi'}$ and variance $\mathrm{Var}(\pi')$ changes.

# D   RETRIEVAL EXAMPLES

We provide retrieval examples of our ProbHash and HashUQ in Figures 10a and 10b using 5 exemplar query images from ImageNet. On the left-most of each row is the query image, and the corresponding retrieved images are plotted on the right-hand-side of the dividing line. We plot each retrieved images relevant to the query in green bounding boxes, and images irrelevant to the query in red bounding boxes. To illustrate the main idea and show the difference between the predicted ordering of the retrieval models with and without uncertainty, here we set number of retrieved samples $r = 10$. (This means that we only retrieve 10 images from all the $128,503$ entries in the database based on the hash-codes Hamming distance for each query, and re-rank these 10 images by the predicted uncertainty of hash-code accordingly.)

By comparing Figure 10a and Figure 10b, we see that our HashUQ will prioritize images with relevant concepts to the query while de-prioritize images with irrelevant concepts, which lead to more principled retrieval ordering. In real case, by rank the data with the same hash-code Hamming distance to the query using the predicted uncertainty, the relevance of top $r$ data to be retrieved will also be improved, and the retrieval accuracy will be affected not only by the retrieval ordering but also in terms of the data to be retrieved.

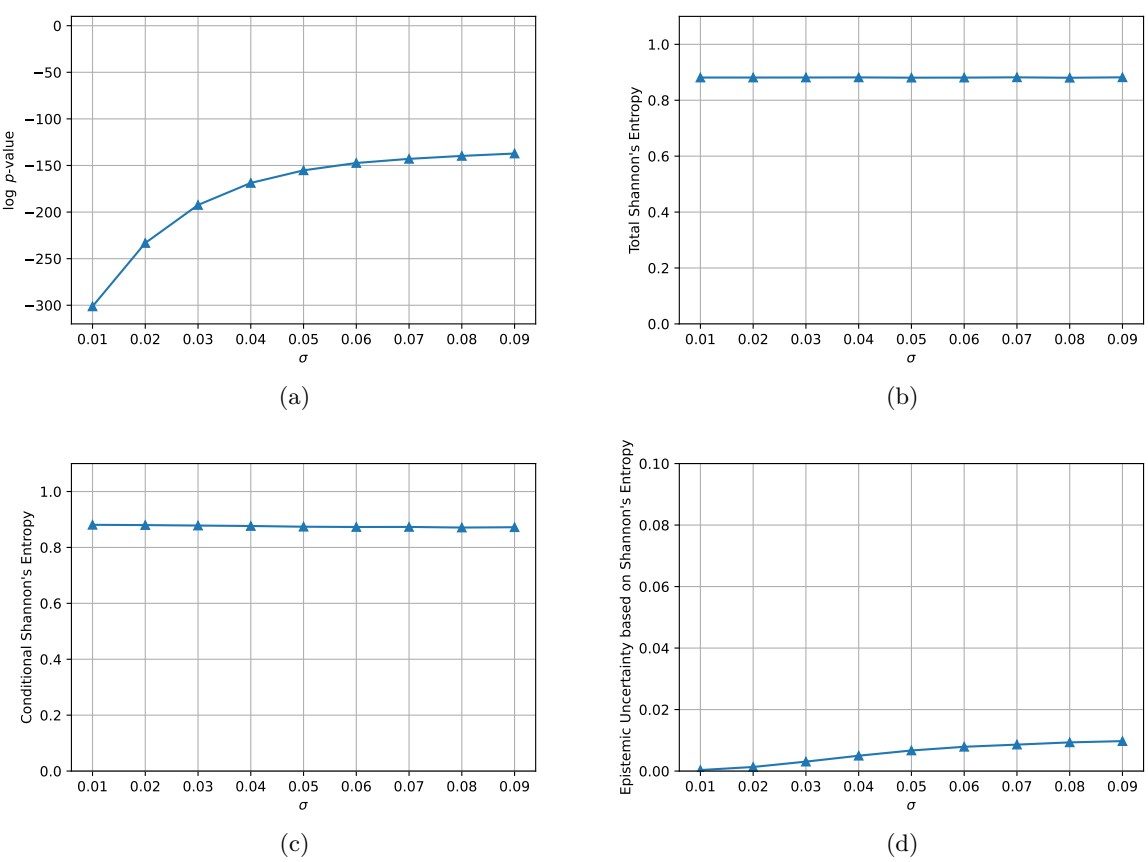

Figure 8: Uncertainty measured with (a) paired sample $t$-test, (b) conditional Shannon's entropy, (c) total Shannon's entropy and (d) induced epistemic uncertainty based on Shannon's entropy using 100 samples from Gaussian distributions $\mathcal{N}(\mu, \sigma^2)$ with respect to standard deviation $\sigma$ changing from 0.01 to 0.1. The mean $\mu = 0.3$ and the Gaussian distribution is truncated at $(a, b)$ with the minimum and maximum value set to be $a = 0.2, b = 0.4$.

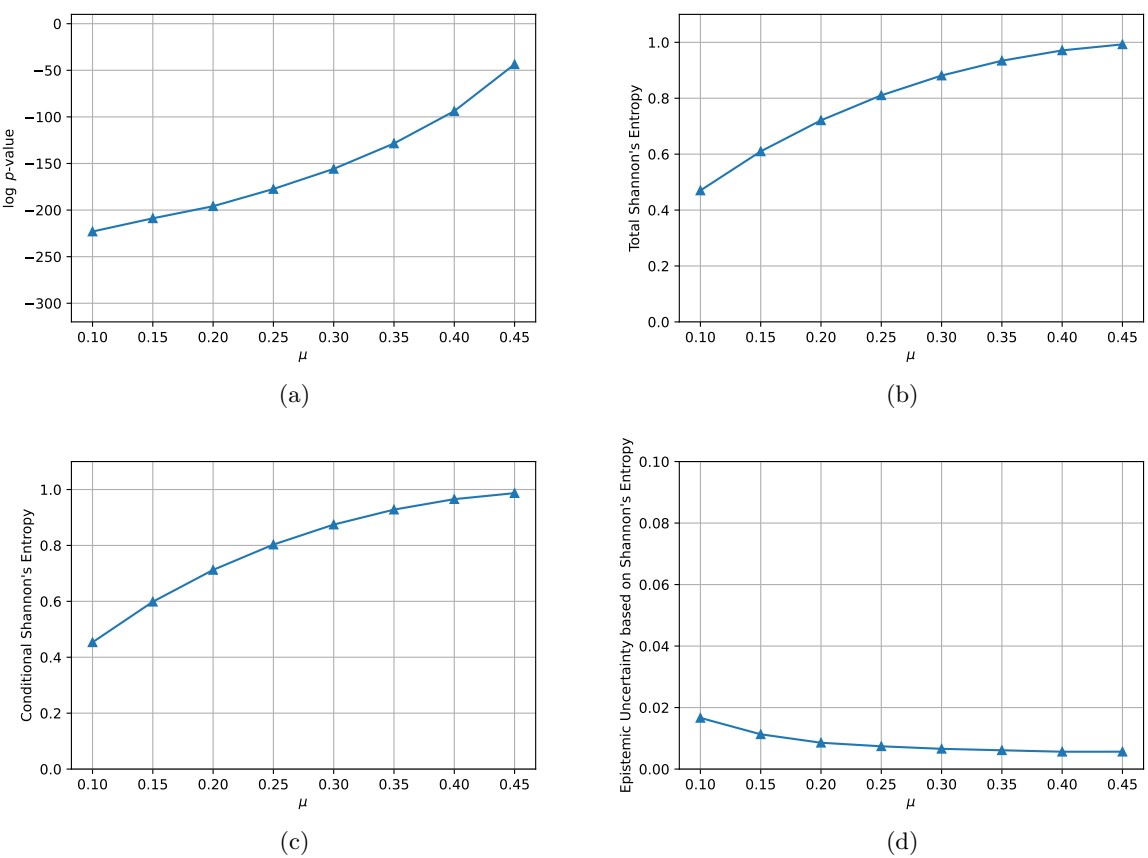

Figure 9: Uncertainty measured with (a) paired sample $t$-test, (b) conditional Shannon's entropy, (c) total Shannon's entropy and (d) induced epistemic uncertainty based on Shannon's entropy using 100 samples from Gaussian distributions $\mathcal{N}(\mu, \sigma^2)$ with respect to mean $\mu$ changing from 0.1 to 0.5. The standard deviation $\sigma = 0.05$ and Gaussian distribution is truncated at $(a, b)$ with the minimum and maximum value $(a, b)$ set to be $a = \mu - 0.1, b = \mu + 0.1$.

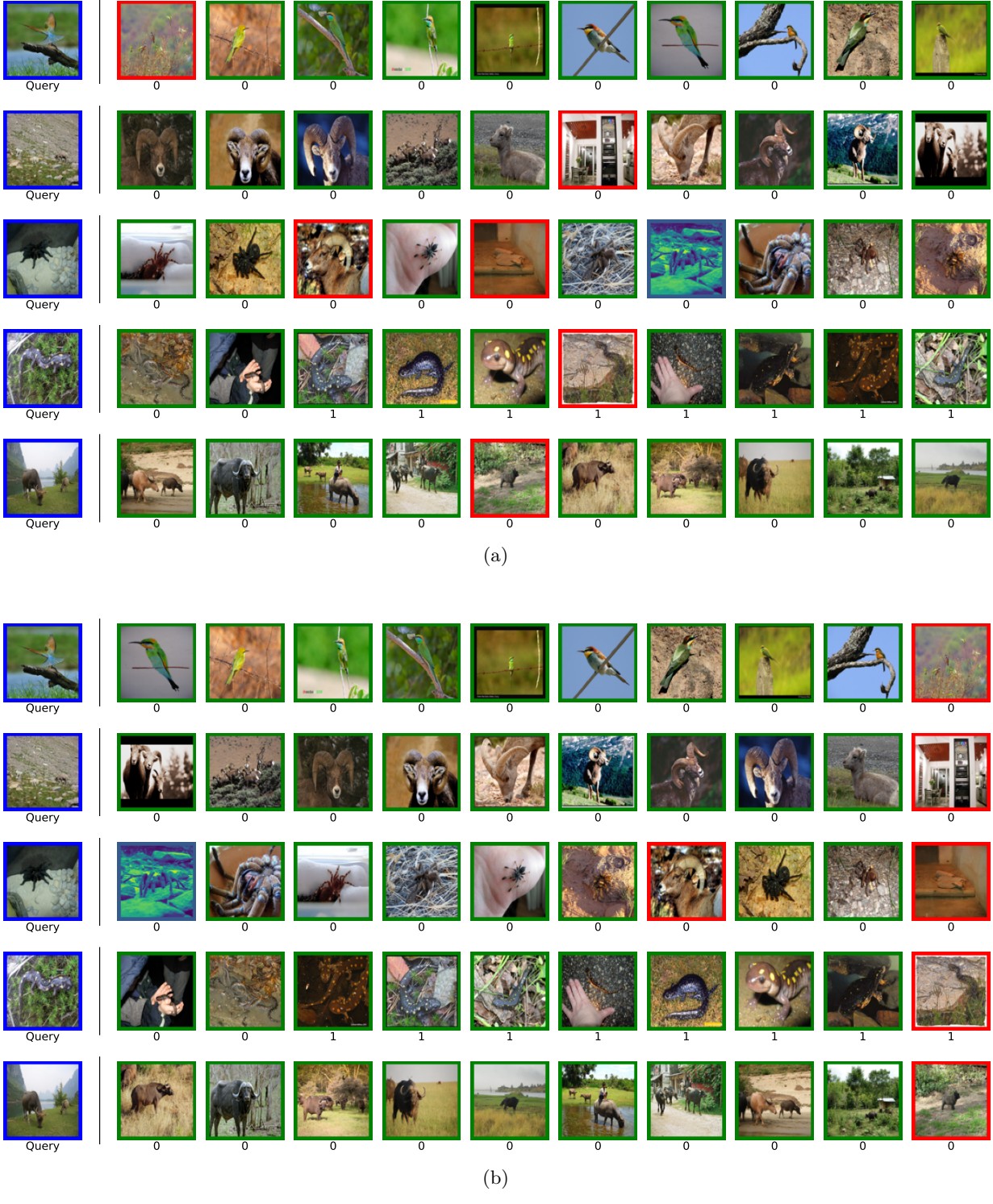

Figure 10: (a): Exemplar image retrieval results using 5 query images from the ImageNet dataset. The retrieval is performed without considering the uncertainty of hash-codes (b) Image retrieval using the same queries with proposed HashUQ. On the left side of the dividing line is the query image while on the right side of the line is the retrieved data, with the decreasing predicted relevance from left to right. We plot success retrievals in green bounding boxes and failure retrievals in red bounding boxes. Below each retrieved image is the Hamming distance of the hash-code to the query image. To illustrate the main idea, here we only retrieve 10 images for each query and re-rank them based on the quantified uncertainty.

# E  SUPPLEMENTARY FIGURES

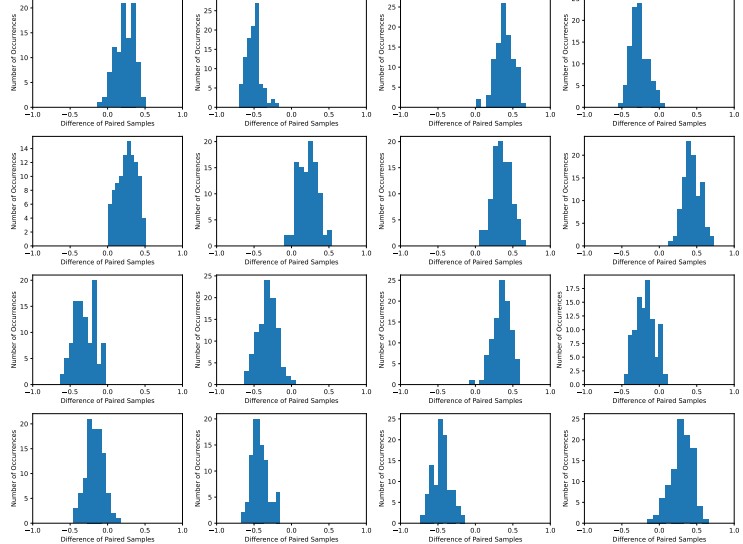

(a)

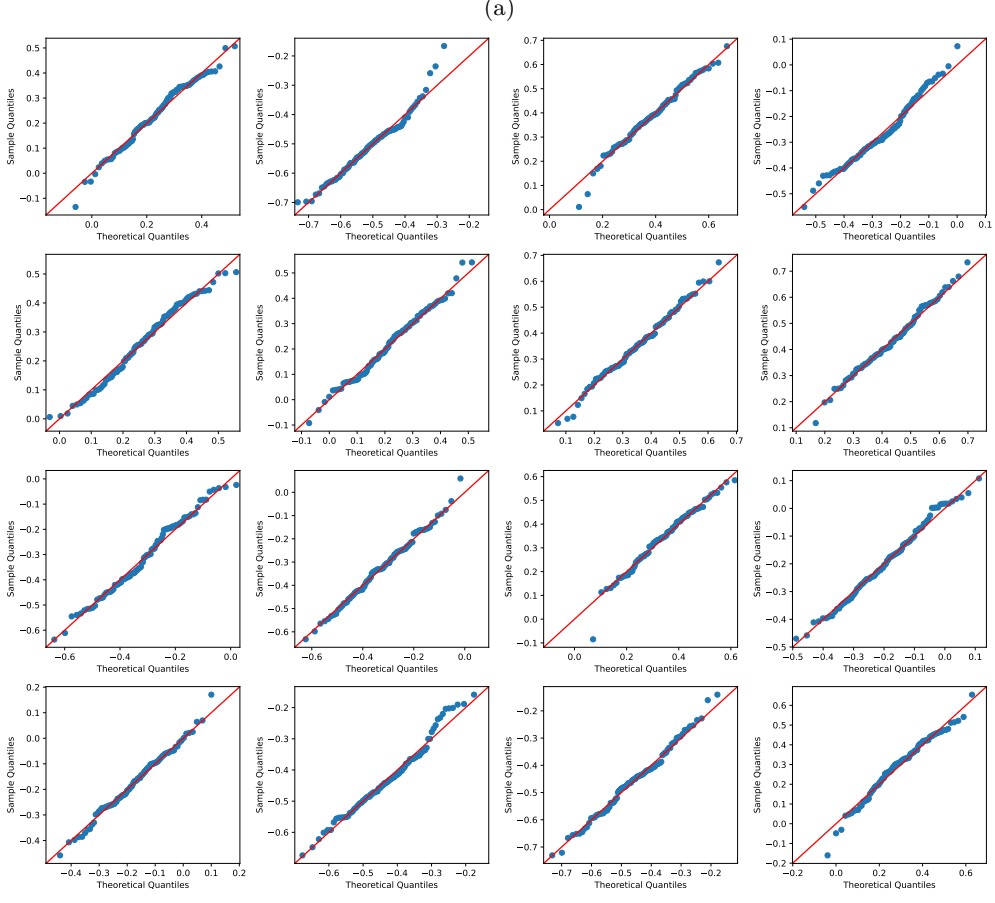

(b)

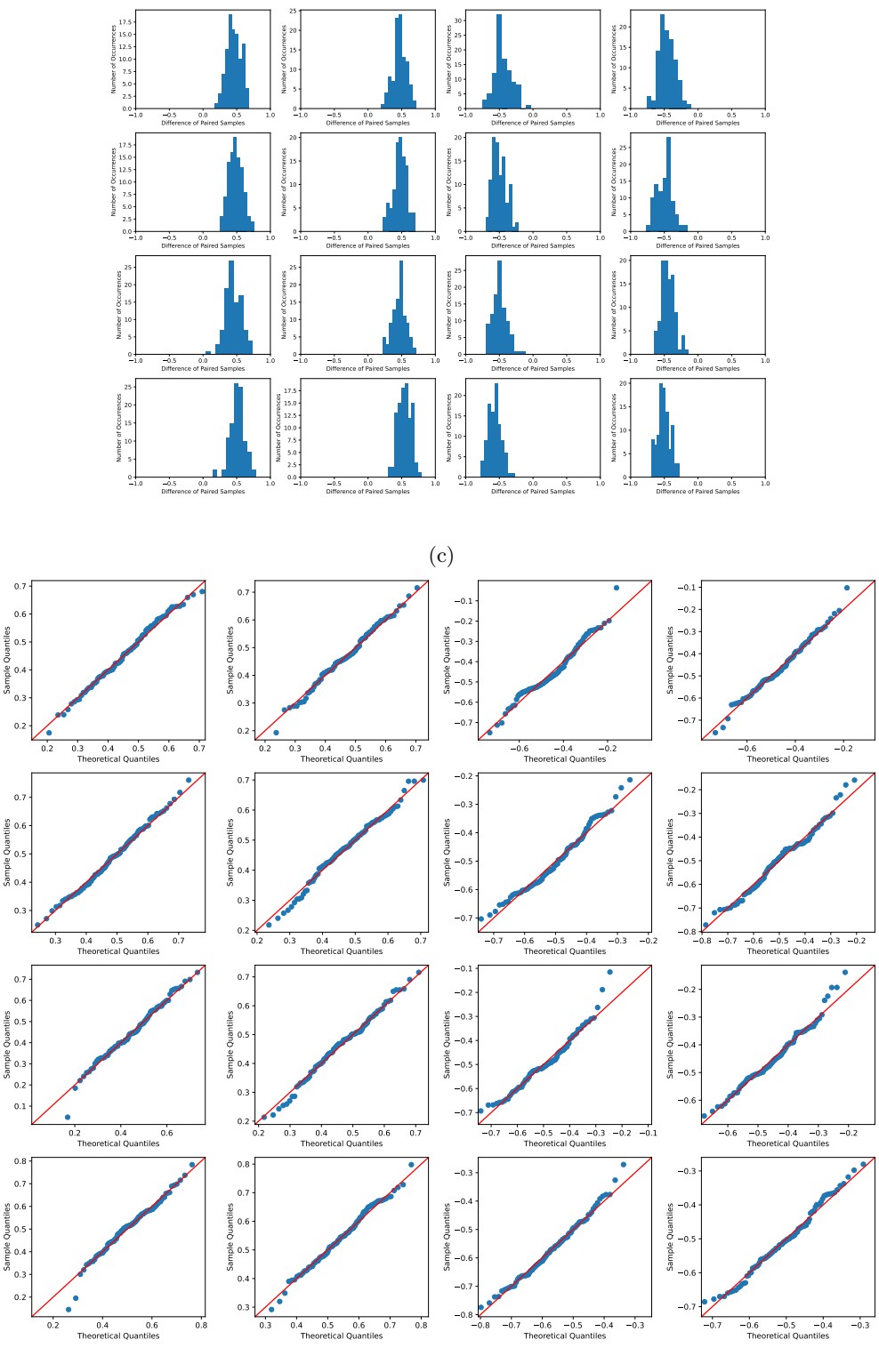

(c)

(d)

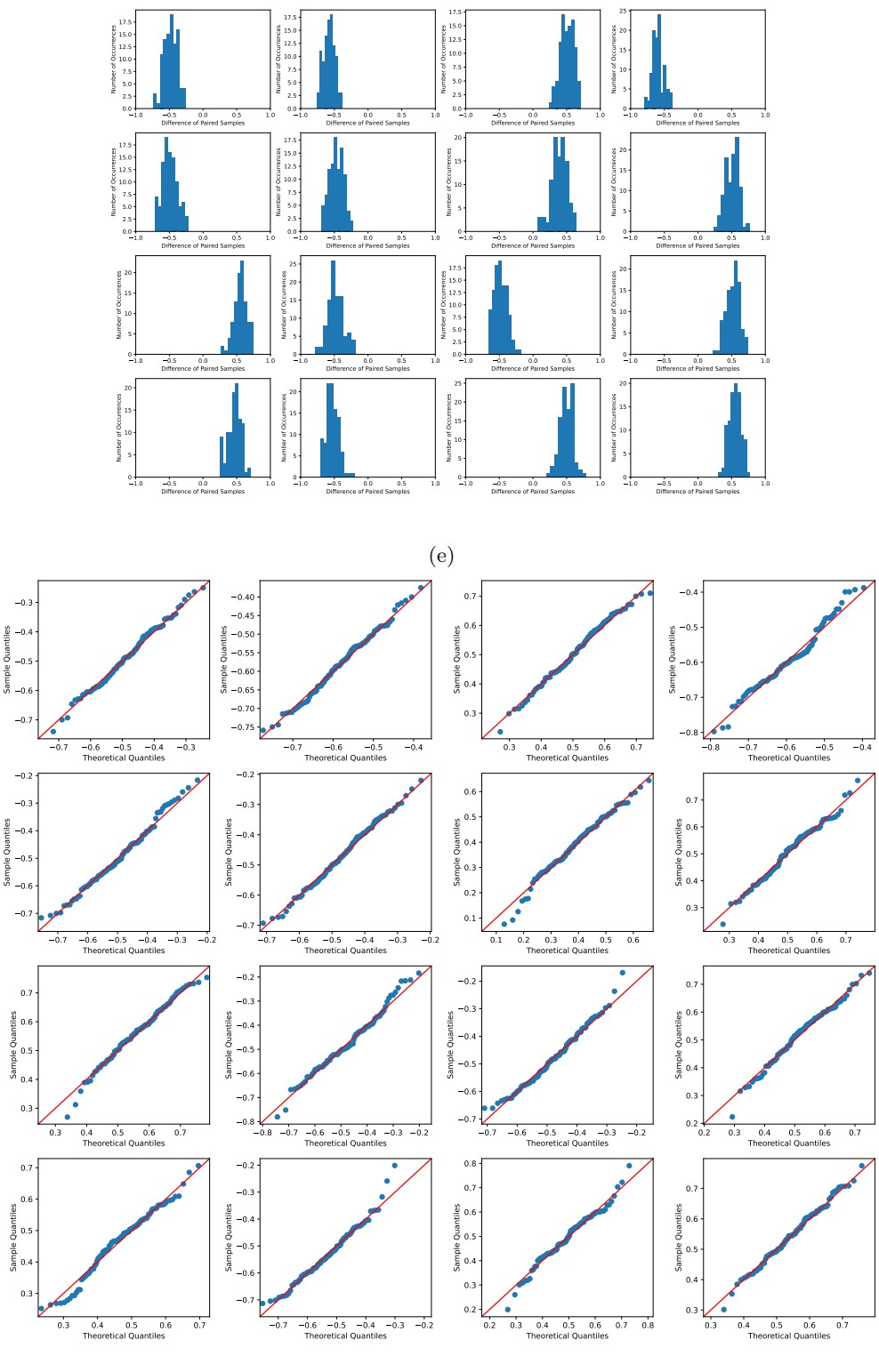

(e)

(f)

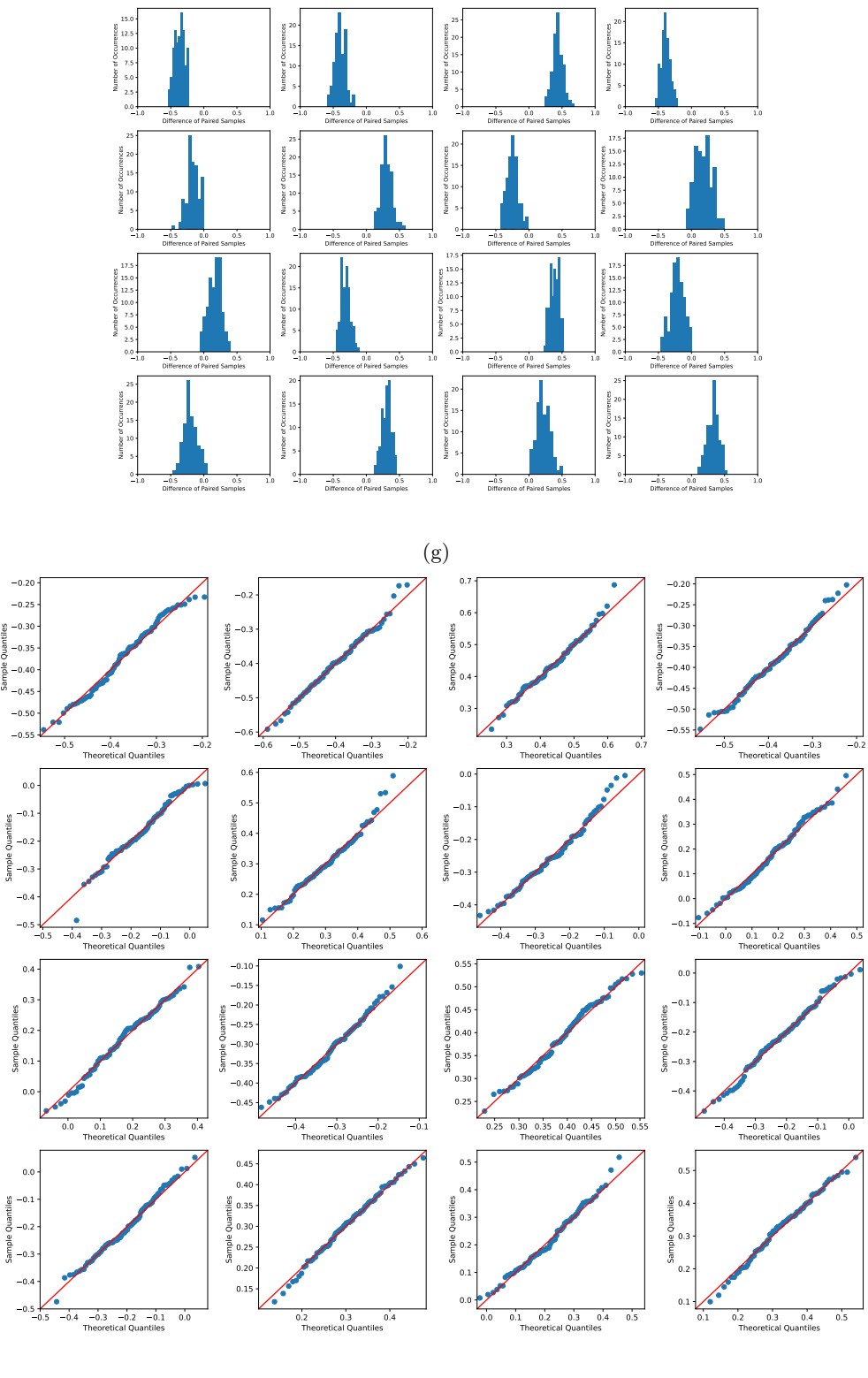

(g)

(h)

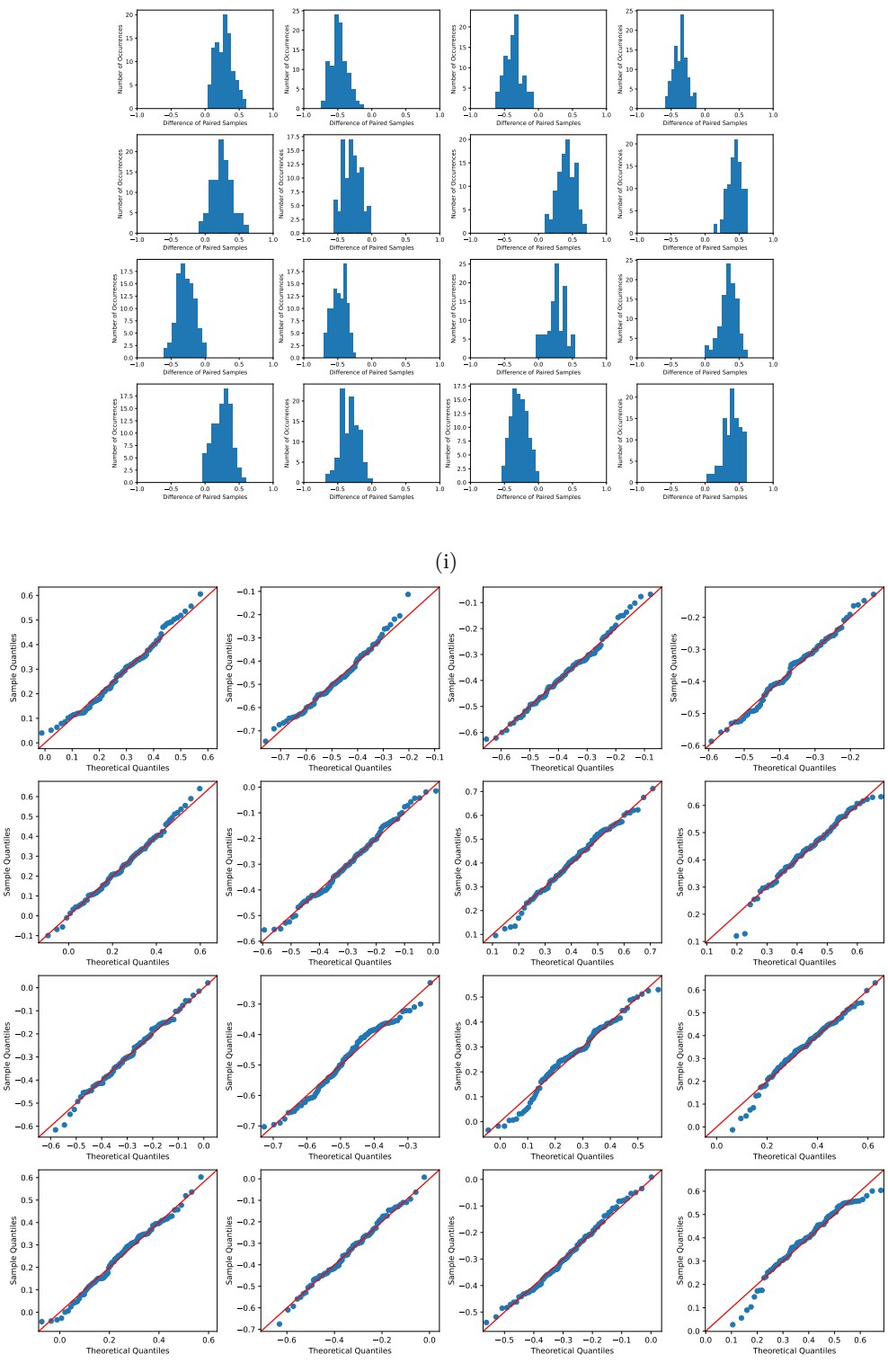

(i)

(j)

Figure 11: Histograms and Q-Q plots of the difference between paired samples of each Bernoulli success rate of hash-code vectors generated using 5 query images from the ImageNet dataset. (a)(c)(e)(g)(i): Histograms of the difference between paired samples. (b)(d)(f)(h)(j) Q-Q plots of the difference between paired samples with respect to the Gaussian distribution with the same mean and variance. We use the same images to generate the plots as in Section D with $K = 16$.

