# OpenReview forum: "Hashing with Uncertainty Quantification via Sampling-based Hypothesis Testing"
_TMLR — Accepted by TMLR_

### Review · Reviewer_3tbV · 2024-07-28

**Summary Of Contributions:**

For the problem of the uncertainty quantification of hash-codes for image retrieval, a novel probabilistic hashing model named ProbHash is proposed to take different sources of uncertainty into account. In addition, an uncertainty-aware hashing strategy named HashUQ is proposed to improve the retrieval performance with little computation and storage overhead. The experimental results on three image datasets validate the effectiveness of the proposed method.

**Audience:**

Yes

**Broader Impact Concerns:**

This work does not seem to have any potential negative societal impact.

**Claims And Evidence:**

Yes

**Requested Changes:**

Please refer to the Weaknesses for details.

**Strengths And Weaknesses:**

Strengths:
1. The proposed method shows that considering the quantified uncertainty is an effective and efficient way to enhance retrieval accuracy.
2. It is natural and intuitive to use statistical hypothesis testing as the measure of the uncertainty quantification of hash-codes.
3. Numerical experiments and analysis demonstrate the effectiveness of considering the quantified uncertainty in improving the retrieval performance.

Weaknesses:
1. The proposed method assumes that the likelihood distribution of each class follows a categorical distribution parameterized by a neural network with the softmax activation and the variational posterior follows a factorized Bernoulli distribution parameterized by a neural network with the sigmoid activation function. However, the rationality of the assumption that these random variables follow various distributions has not been explained.
2. The rationality of using neural networks to model the parameters of these distributions is not discussed and analyzed, which makes the proposed method not convincing enough. In addition, can the selection of different network structures significantly affect the retrieval performance of the final model? Such problems require further numerical experiments, analysis and discussion. At least some empirical discussion and explanation about the selection of different network structures are necessary.
3. To derive ProbHash for existing “Center-Target” methods, the likelihood distribution is assumed to be proportional to the exponential of the Hamming distance, the rationality of this assumption needs to be further discussed and analyzed since the lack of theoretical analysis on the consistency.

---

> ### Author Response · Authors · 2024-08-31
> **Author Response to Reviewer 3tbV**
>
> We appreciate the reviewer for recognizing the effectiveness of the proposed method and strength in numerical experiments. We are addressing the weaknesses raised by the reviewer as follows:
>
> **Question 1: The proposed method assumes that the likelihood distribution of each class follows a categorical distribution parameterized by a neural network with the softmax activation and the variational posterior follows a factorized Bernoulli distribution parameterized by a neural network with the sigmoid activation function. However, the rationality of the assumption that these random variables follow various distributions has not been explained.**
>
> The image classes are discrete variables and the hash-codes are binary variables in our supervised image hashing setup. It is the convention and norm to use the categorical distribution for discrete random variables and Bernoulli distribution for binary random variables. Different methods mainly differ on how the probability or the corresponding logits are modeled. One main difference between our work and previous works is the modeling of latent hash-codes distribution: previous works model the hash-codes as Bernoulli variables with the success probabilities always 1 or 0 by thresholding the sigmoid activation output of a neural network. One important contribution of our work is to model the hash-codes as Bernoulli random variables with the logits directly parameterized by a neural network, which does not constrain the success probabilities of Bernoulli variables with the thresholding function, leading to end-to-end learning with our gradient estimates. All these help improve the model flexibility and enable the uncertainty quantification capability.
>
> **Question 2: The rationality of using neural networks to model the parameters of these distributions is not discussed and analyzed, which makes the proposed method not convincing enough. In addition, can the selection of different network structures significantly affect the retrieval performance of the final model? Such problems require further numerical experiments, analysis and discussion. At least some empirical discussion and explanation about the selection of different network structures are necessary.**
>
> The neural networks $f_{\boldsymbol\phi}, g_{\boldsymbol\theta}$ are used to capture the dependency between input data and Bernoulli logit parameters, and the dependency between Bernoulli probability and class labels. This is also called *amortized variational inference* [6] and has been adopted in many previous works [7,8].
>
> The network structure should be adapted to the input and output data requirement to introduce different inductive bias, such as the convolutional neural network for images, and transformer for sequential data. Typically a larger network will have more capacity to capture complex dependencies between variables and be more flexible to model complicated probability distribution. Those factors will indeed affect the retrieval performance of the final model.
>
> The reason to select the AlexNet architecture is mainly for the popularity in previous works [1-5], which makes it easier for fair comparison and benchmarking. We additionally test our ProbHash and HashUQ with VGG16 backnone architecture. With deeper structure and better flexibility, the VGG16 can consistently achieve better retrieval accuracy than AlexNet with different lengths of hash-codes, which verifies that the network structure will indeed affect the retrieval accuracy. With the quantified uncertainty integrated into the retrieval, our HashUQ can similarly improve the retrieval performance.
>
> | Method    | 16 bit | 32 bit    | 64 bit    |
> |---|---|---|---|
> | ProbHash-AlexNet   | 0.606    | 0.680    | 0.706    |
> | HashUQ-AlexNet   | 0.631    | 0.696    | 0.718    |
> | ProbHash-VGG16   | 0.792    | 0.833    | 0.846    |
> | HashUQ-VGG16 | 0.810 | 0.846    | 0.857    |

---

> > ### Author Response · Authors · 2024-08-31
> > **Author Response to Reviewer 3tbV**
> >
> > **Question 3: To derive ProbHash for existing “Center-Target” methods, the likelihood distribution is assumed to be proportional to the exponential of the Hamming distance, the rationality of this assumption needs to be further discussed and analyzed since the lack of theoretical analysis on the consistency.**
> >
> >
> > For a given data sample, the hash-code center corresponding to its class-label will be *more likely* to have small Hamming distance to the unobserved hash-codes compared to large distance. We assume this probability to decay exponentially as the Hamming distance getting larger by assuming likelihood proportional to the exponential of the negative Hamming distance, which has also been adopted in previous works [1]. Another reason for the choice of this specific likelihood function is that we can derive an closed-form optimization objective without resorting to the biased straight-through or Gumbel-Softmax gradient estimators to approximate the gradient of the undifferentiable Bernoulli variables. We acknowledge the existence of other choices of likelihood functions to achieve the similar property, and defend our selection for the aforementioned benefits.
> >
> > References:
> >
> > [1] Yuan, Li, et al. "Central similarity quantization for efficient image and video retrieval." Proceedings of the IEEE/CVF conference on computer vision and pattern recognition. 2020.
> >
> > [2] Cao, Zhangjie, et al. "Hashnet: Deep learning to hash by continuation." Proceedings of the IEEE international conference on computer vision. 2017.
> >
> > [3] Su, Shupeng, et al. "Greedy hash: Towards fast optimization for accurate hash coding in cnn." Advances in neural information processing systems 31 (2018).
> >
> > [4] Hoe, Jiun Tian, et al. "One loss for all: Deep hashing with a single cosine similarity based learning objective." Advances in Neural Information Processing Systems 34 (2021): 24286-24298.
> >
> > [5] Doan, Khoa D., Peng Yang, and Ping Li. "One loss for quantization: Deep hashing with discrete wasserstein distributional matching." Proceedings of the IEEE/CVF Conference on Computer Vision and Pattern Recognition. 2022.
> >
> > [6] Gershman, Samuel, and Noah Goodman. "Amortized inference in probabilistic reasoning." Proceedings of the annual meeting of the cognitive science society. Vol. 36. No. 36. 2014.
> >
> > [7] Kingma, Diederik P., and Max Welling. "Auto-encoding variational bayes." arXiv preprint arXiv:1312.6114 (2013).
> >
> > [8] Fan, Xinjie, et al. "Bayesian attention modules." Advances in Neural Information Processing Systems 33 (2020): 16362-16376.

---

> > > ### Comment · Reviewer_3tbV · 2024-09-18
> > >
> > > The authors' response partially addressed my concerns. However, the response to the third question is not convincing enough. The use in previous work cannot explain the rationality of this assumption. The consideration for optimization purposes can only partially but not essentially explain the rationality of this assumption. Since the reasons provided by the authors cannot replace the role of theoretical analysis on the consistency, this assumption is almost heuristic to a certain extent. The authors need to provide some specific discussions and analyses in the revised version, such as experimental comparisons with other likelihood functions or more insightful discussions and analyses.

---

> ### Author Response · Authors · 2024-09-18
> **Author Response to Reviewer 3tbV**
>
> We thank the reviewer for additional discussion.
>
> The likelihood function is a model choice when constructing the probabilistic machine learning model. *In Bayesian statistics, the likelihood represents our beliefs about what data we expect to see for each setting of parameters of the model*[1][2]. The choice of likelihood distribution will depend on the data type as well as the convenience of computation (For example, people prefer conjugate prior-likelihood pairs for easy posterior inference). The likelihood function should take the higher value when data is consistent with the model while the lower value when the data is inconsistent with the model. For example, the *isotropic Gaussian distribution* is typically chosen for continuous data, and the *categorical distribution*  is typically used for modeling the discrete data which can possibly take $M$ different values, $M \geq 2$. Both belong to the exponential family and have easy-to-compute log-likelihood function.
>
> The data we are modeling is binary vectors of dimension $K$. We choose the likelihood distribution:
>
> $$p(c_i|b_i) \\propto \\exp(- \\phi D_{Hamming}(c_i, b_i))$$
>
> for the following reasons:
> - *This probability distribution matches with our aforementioned principle for the likelihood choice:* Both $c_i$ and $b_i$ are binary vectors and the Hamming distance is indeed a proper choice of distance measure for binary vectors. The adopted likelihood distribution reflects our belief that the hash-code center $c_i$ corresponding to its class-label will be more likely to have small Hamming distance to the unobserved hash-code $b_i$. Our adopted likelihood distribution takes the highest value when $c_i$ match with $b_i$ and decrease when $c_i$ gradually deviate from $b_i$.
> - *This probability distribution belongs to the commonly adopted likelihood distributions:* Just like Gaussian or categorical distribution, our adopted likelihood distribution is a Boltzmann distribution, which is one of the most widely used probability distribution in statistical analysis and machine learning modeling. Our adopted distribution also belongs to the exponential family, which has a simple form of log likelihood function.
> - *This probability distribution has computational advantages over other choices of likelihood distributions:*  This specific choice of our likelihood distribution will also lead to the closed-form optimization objective as emphasized in our manuscript. The benefit of this closed-form objective is two-fold: (1) it reduces the computational consumption for conducting multiple forward Monte Carlo samples; (2) it is unbiased and will reduce the variance for stochastic gradient descent, which will lead to more efficient model training.
>
> We have included a section discussing the choice of our likelihood function in *Appendix A.3 of our updated manuscript*, and we are willing to compare our chosen likelihood with alternative ones the reviewer may help suggest.
>
> [1] Murphy, Kevin P. Probabilistic machine learning: Advanced topics. MIT press, 2023.
>
> [2] Barber, David. Bayesian reasoning and machine learning. Cambridge University Press, 2012.

---

> > ### Comment · Reviewer_3tbV · 2024-09-20
> >
> > The authors' response has addressed my concern about the rationality of the assumption on the likelihood distribution. Further, I would like to know whether the selected Boltzmann distribution is optimal or general, and whether other commonly used distributions will affect the performance of the proposed method. I expect the authors to provide some comparative experiments with other distributions as much as possible, so that this model selection problem will be well handled and the completeness of the proposed method can be improved. I am satisfied with the authors' handling of other model selection problems involved in the proposed method.

---

> ### Author Response · Authors · 2024-09-22
> **Author Response to Reviewer 3tbV**
>
> We appreciate the reviewer for additional comments, and we are glad that the reviewer is satisfied with the rationality of the likelihood distribution and the handling of other model selection problems. Here we try to explain why other commonly used distributions may not be usable in our problem; and we also provide another likelihood distribution to compare with the distribution adopted in our paper.
>
> Many other commonly used likelihood distributions are not directly applicable to our problem since:
> - The *isotropic Gaussian distribution*:
>
> $$p(c_{i, k}|b_{i,k}) = \\frac{1}{\\sqrt{2\\pi\\sigma_k^2}}\\exp(-\\frac{(c_{i,k} - b_{i,k})^2}{2\\sigma_k^2}) $$
>
> requires $c$ to be a continuous random vector.
> - The *categorical distribution*:
>
> $$p(c\_{i, k}|b\_{i,k}) = b\_{i,k}^{c\_{i, k}}(1 - b\_{i,k})\^{(1 - c\_{i, k})}$$
>
> requires $0 < b_{i,k} < 1$.
>
> We thus try to compare our adopted likelihood distribution with another Boltzmann distribution:
>
> $$p(c_i|b_i) \propto \exp(- \phi D^2_{Hamming}(c_i, b_i))$$
>
> This distribution takes a similar format as Gaussian distribution, and the closed-form log-likelihood can also be derived as:
>
> $$\\mathbb{E}_{q\\left(\boldsymbol{b}_i|\\boldsymbol{x}_i, \\boldsymbol{\\psi}\\right)}\\left[-\\log p(\\boldsymbol{c_i}|\\boldsymbol{b_i})\\right] $$
>
> $$ = \\mathbb{E}_{q\\left(\\boldsymbol{b}_i|\\boldsymbol{x}_i, \\boldsymbol{\\psi}\\right)}\\left[\\phi|\\boldsymbol{c}_i - \\boldsymbol{b}_i|^2 - \\log Z\\right]$$
>
> $$ = \\mathbb{E}\_{q\\left(\\boldsymbol{b}\_i|\\boldsymbol{x}\_i, \\boldsymbol{\\psi}\\right)}\\left[\\phi\\sum_k\^K|c_{i,k} - b_{i,k}|^2 - \\log Z\\right]$$
>
> $$ = \\phi\\sum\_k^K\\left[ c\_{i,k}\^2 + 2c\_{i,k}\ + 1 - 4 c\_{i,k} {\\boldsymbol\\sigma}\\left(\\boldsymbol{f}\_{\\boldsymbol{\\psi}, k}\\left(\\boldsymbol{x}\_i\\right)\right) \^2 \\right] - \\log Z.$$
>
> This means that this distribution also satisfies all the three principles for selecting likelihood function mentioned in the previous comment. The main difference between the distribution above and the likelihood distribution we adopt in the main paper is how the probability decays as the Hamming distance gets larger. Notice that there may exist other distributions of the form $p(c_i|b_i) \\propto \\exp(- \\phi D\^{\\beta}\_{Hamming}(c\_i, b\_i))$ which can satisfy the above properties, but the closed-form optimization objective will become complicated or even intractable when a large $\beta$ is used. We run additional experiments on the ImageNet dataset to compare the distribution above ($D^2_{Hamming}$) and the likelihood distribution we adopt in the main paper($D_{Hamming}$), and include the results in the table below:
>
> | Method    | 16 bit | 32 bit    | 64 bit    |
> |---|---|---|---|
> | ProbHash-$D_{Hamming}$   | 0.606    | 0.680    | 0.706    |
> | HashUQ-$D_{Hamming}$    | 0.631    | 0.696    | 0.718    |
> | ProbHash-$D^2_{Hamming}$    | 0.606    | 0.678 | 0.705    |
> | HashUQ-$D^2_{Hamming}$    | 0.631    | 0.692   | 0.716    |
>
> We observe that the retrieval accuracy is slightly affected with the likelihood distribution in the main paper replaced with the likelihood discussed above. The experimental results also show that our HashUQ strategy can help improve the retrieval accuracy regardless of the likelihood choice. We have included these experimental results along with the discussion about the rationality on the likelihood choice in *Appendix A.3*.

---

> > ### Comment · Reviewer_3tbV · 2024-09-24
> >
> > I am of course aware that the isotropic Gaussian distribution and the categorical distribution are not applicable to the proposed method, and my intention was to try to get the authors to explore the range of applicable likelihood distributions as much as possible. The authors' response actually provides a rough criterion for the selection of the likelihood distribution, i.e., a simple closed-form log-likelihood. To some extent, the selection of the likelihood distribution is specific to the proposed method, but the range of applicable likelihood distributions is still promising. Overall, the authors have addressed my concerns and I will recommend acceptance.

---

### Review · Reviewer_Qmeu · 2024-08-09

**Summary Of Contributions:**

The authors consider using neural networks to learn efficient hash functions for information retrieval using labelled data (specifically labelled images in the paper). The authors propose a variational information bottleneck architecture: the encoder network maps data to binary vector-valued latent random variables, which serve as the basis for the hash of the data. Then, during training, a separate decoder network aims to reconstruct the label or a proxy for the label; the authors specifically map the labels to certain error-correcting code vectors in the hash-space, simplifying the decoding and training procedure.

The authors propose using Monte Carlo dropout (leaving dropout turned on during inference time and averaging results over multiple forward passes through the network) to quantify the epistemic uncertainty in their network's prediction. Furthermore, they develop an information retrieval procedure that considers the predicted uncertainty, which ranks data that map to the same hash based on the model's confidence in the prediction. The authors also account for their method's storage requirements and experiment with different quantization levels for the predicted uncertainty, which they save along the hash of each data point.

They empirically verify the proposed approach's efficiency on three image datasets, and their method is competitive with state-of-the-art ones. Moreover, they conduct several ablation studies to show the value brought by each new component they propose.

**Audience:**

Yes

**Claims And Evidence:**

No

**Requested Changes:**

## Necessary:
 - Please address the concerns I outline above, and update the paper accordingly.
 - Additionally, please increase the line thickness in the Figures, as they are currently hard to read on a laptop screen without zooming in.

## Nice to have:
 - The presentation of the "label-target" and "center-target" variants of the method are not separated cleanly enough; please signpost them better using bold paragraph headings or putting them in different subsections. I would actually suggest that the authors present only a generic framework and specialize the algorithm at the end of the section.

## Typos/style:
- introduction: "uncertainty-awared" -> "uncertainty aware"
- Please format the argmax above section 4 so that the optimization variable is below the word "argmax" and not to the side.
 - notation: $x^q$ clashing with $x^k$,  please use a double subscript $x_{i,k}$ instead
 - first sentence of section 3.4: "$x$ and $\psi$ and $p(\psi)$" - writing $\psi$ not needed.
 - In many equations (e.g. eq (1)): please use `\left[` and `\right]` for the brackets.

**Strengths And Weaknesses:**

## Strengths
The paper contains two strong ideas:
- Using a variational distribution over the hashes, which, in the "center-target" variant of the method, allows the authors to analytically evaluate the model's expected likelihood. Therefore, the gradient of the expectation is also available exactly, and there is no need for a straight-through estimator, which is both biased and has a higher variance.
 - Using uncertainty estimates to augment information retrieval.

I also found the experimental evaluation and ablation studies mostly satisfactory. I appreciated that the authors paid attention to their method's increased storage requirements and, therefore, attempted to make fair comparisons with competing methods.

## Weaknesses
Methods-wise, I have three main concerns:
- The authors do not compare Monte Carlo dropout to 1) only using dropout to regularise training or 2) to a fully-factorized Gaussian weight posterior. These comparisons would be important for two reasons: 1) doing variational inference is generally more expensive than using point estimation, so it would be good to establish whether the extra computation is worth it, and 2) Monte Carlo dropout as a variational approximation has well-known shortcomings for uncertainty quantification (see, e.g. [1]). Hence, it would be good to see if using a better approximation could be used to improve performance.
 - There is no motivation for using the t-test as the basis for uncertainty quantification; measuring entropy instead seems a lot more sensible. Based on the paper's contents, my impression is that the authors chose it because it performed slightly better than other measures in the experiments they conducted (though the statistical significance is unclear). One sentence I found particularly problematic was the authors saying, "We find [hypothesis testing] to be especially suitable for quantifying the uncertainties of hash-codes considering the nature of the problem." - there is no explanation of what particular nature of the problem should make it suitable.
 - There isn't a clear motivation for the authors' ranking criterion. While using the uncertainty values as a secondary ordering for data whose hashes collide is certainly sensible, it also seems arbitrary. Could the authors provide a principled justification for why ordering data this way should work better in some cases than something else? Furthermore, one dissatisfactory thing about the authors' method is that the training and the test/retrieval procedures seem to differ. Concretely, the authors use the uncertainty values to extend the hashes of the data in the pool set but don't use them for the query point. Could the authors comment on this?

Content-wise, the method appears to be an instance of the variational bottleneck framework [2], but the authors do not discuss this in the related works section. Could the authors comment on this?

Finally, the paper's presentation needs improvement. Many statements are imprecise, have information missing or are potentially misleading:
 - In Section 3, when presenting the necessary background, the authors note that the methods they base their work on use the " binary quantization function $\mathrm{sgn}$". The standard definition of this function maps to $\{-1, 1\}$. The next paragraph mentions that they generate the centroids for their method's "center-target" variant using a Hadamard matrix. Firstly, the authors should define what they mean by the Hadamard matrix in the paper, as I don't think it is a widely-known object. Furthermore, there are two conventions for these matrices, one which defines that the entries are $\{-1, 1\}$ (the one I was familiar with) and $\{0, 1\}$. However, the authors adopt the $\{0, 1\}$ convention in the rest of the paper, so I found the whole method quite confusing on the first couple of reads; please update the paper to avoid this confusion.
 - The main text does not define the performance metric the authors use in the experiments (mAP@1000).
 - Table 2 uses unnecessary abbreviations (Opt, ST, CF) that are not defined in the table caption. Please just spell out the relevant phrases.
 - The authors below Eq (2) state, "λ is a hyperparameter to reflect our belief of the data relative to the prior." I don't understand what this should mean; could the authors please clarify? On a related note, do the authors tune the prior parameters?
 - The related works section (Section 2) is essentially just a literature review that does not specify how the works mentioned are related to the authors' work or how they differ; please clarify this. Furthermore, I found this section quite dense straight after the introduction; I suggest the authors move it after the experiments section.

## References
 - [1] Foong, A., Burt, D., Li, Y., & Turner, R. (2020). On the expressiveness of approximate inference in bayesian neural networks. Advances in Neural Information Processing Systems, 33, 15897-15908.
 - [2] Alemi, A. A., Fischer, I., Dillon, J. V., & Murphy, K. (2016). Deep variational information bottleneck. arXiv preprint arXiv:1612.00410.

---

> ### Author Response · Authors · 2024-08-31
> **Author Response to Reviewer Qmeu**
>
> We would like to express our gratitude to the reviewer for the valuable and constructive suggestions. Here we would like to provide our response to the major points raised by the reviewer:
>
> **Question 1: The authors do not compare Monte Carlo dropout to 1) only using dropout to regularise training or 2) to a fully-factorized Gaussian weight posterior. These comparisons would be important for two reasons: 1) doing variational inference is generally more expensive than using point estimation, so it would be good to establish whether the extra computation is worth it, and 2) Monte Carlo dropout as a variational approximation has well-known shortcomings for uncertainty quantification (see, e.g. [1]). Hence, it would be good to see if using a better approximation could be used to improve performance.**
>
>
> We applied dropout to regularize both the baseline and ProbHash models during training, as detailed in the experiments of our original manuscript. We have updated *Appendix* B.1.2 of our manuscript and clearly mentioned this to avoid ambiguity.
>
> Here we therefore additionally include our ProbHash and HashUQ with (1) deterministic neural network weights (2) fully factorized Gaussian weights with reparameterization VI(Bayes-by-Backprop). The empirical evaluation of our ProbHash and HashUQ with determinstic weights(Determinstic), Fully Factorized Gaussian weight(FFG) and dropout VI(MCD) are shown in the table below.
>
> | Method    | 16 bit | 32 bit    | 64 bit    |
> |---|---|---|---|
> | ProbHash-FFG | 0.597 | 0.647    | 0.662    |
> | ProbHash-MCD   | 0.606    | 0.680    | 0.706    |
> | ProbHash-Deterministic | 0.546     | 0.642    | 0.702    |
> | HashUQ-FFG | 0.633 | 0.677    | 0.686    |
> | HashUQ-FFG-Bin | 0.628 | 0.673    | 0.678    |
> | HashUQ-MCD   | 0.631    | 0.696    | 0.718    |
> | HashUQ-MCD-Bin   | 0.623    | 0.690    | 0.714    |
>
> The quantified uncertainties of Fully Factorized Gaussian weights can similarly be used to help enhancing the retrievals as the dropout weights.
> We observe worse empirical performance of Fully Factorized Gaussian weights compared to dropout in most of the setups with or without using uncertainties. In the meanwhile, the performance improvement of uncertainties quantified from the Fully Factorized Gaussian weights is slightly more significant compared to dropout. While some more flexible distribution families may approximate the posterior and model the uncertainties better, it also brings challenges to the optimization, which may require further analysis on the ELBO maximization objective and gradient based optimization methods.
> We include the full experiment details along with the discussion about the results in the appendix B.5 of our updated manuscripts.

---

> > ### Author Response · Authors · 2024-08-31
> > **Author Response to Reviewer Qmeu**
> >
> > **Question 2: There is no motivation for using the t-test as the basis for uncertainty quantification; measuring entropy instead seems a lot more sensible. Based on the paper's contents, my impression is that the authors chose it because it performed slightly better than other measures in the experiments they conducted (though the statistical significance is unclear). One sentence I found particularly problematic was the authors saying, "We find [hypothesis testing] to be especially suitable for quantifying the uncertainties of hash-codes considering the nature of the problem." - there is no explanation of what particular nature of the problem should make it suitable.**
> >
> > We model both of the neural network weights and hash-codes given input image and network realization as random variables, which gives us two different uncertainties on hand:
> >
> > 1. The uncertainty of hash-codes probability induced by the uncertainty of network weights.
> > 2. The uncertainty of hash-codes given the network realization.
> >
> > While each of these two uncertainty can be quantified separately using either variance or Shannon's entropy, we can unify these two types of uncertainty by performing the student's $t$-test. A justification of the proposed $t$-test based uncertainty measurement, which can also be found in Section 4.1 and 4.2 of our manuscript, is provided below:
> >
> >
> > To clarify the scenario being tested, consider a situation with a binary label for each input $i$, where the probabilities sum to unity: $P(b_i=1) + P(b_i=-1) = 1$ under each random sample. Over $N$ random samples, we can define two label-specific empirical distributions: $$\pi' = \sum_{i=1}^N \frac{1}{N} \delta_{\pi=P(b_i=1)}$$ and $$\pi'' = \sum_{i=1}^N \frac{1}{N} \delta_{\pi=P(b_i=-1)}$$ We aim to assess whether the unknown population means of these two distributions are statistically equivalent, utilizing a two-sample $t$-test for this purpose, and use the corresponding $t$-statistics or $p$-value to measure uncertainty.
> >
> > Consider four different cases when the mean value of ${\pi}\_{i,k}$ to be either $0.6$ or $0.9$, with either low or high variance.
> > If a model consistently predicts $\pi_{i,k} \approx 0.9$, this indicates the model clearly towards predicting $b_{i,k}$ to be ''1''.
> > In either cases when predicting $0.6$ consistently or $0.9$ on average with high variance, the model is still in favor of deciding on ''1'' but with more uncertainty. In the last case when the model outputs with the mean value of $\pi_{i,k}$ to be $0.6$ with high variance, the model is less confidence for the decision on ''1''. These uncertainties can be quantified by conducting student's $t$-test of the null hypothesis that the mean probabilities of being ''1'' and ''-1'' are equal on each of the $k^{th}$ entry. We represent the uncertainty in $b_{i,k}$ using the $p$-value of the test. A smaller $p$-value in favor of rejecting the null hypothesis indicates a more significant difference between the Bernoulli success and failure rates, reflecting our confidence in $b_{i,k}$.
> >
> > The Shannon's Entropy, instead, will ignore the uncertainty induced by network weights, or measure the uncertainty averaged over different network weight realization. We consider student's $t$-test a reasonable uncertainty measure in addition to the Shannon's Entropy and variance, and to be possibly more suitable when these two different uncertainties together presented. We have adjusted our claim in Section 3.3 for better clarity and coherency.

---

> > > ### Author Response · Authors · 2024-08-31
> > > **Author Response to Reviewer Qmeu**
> > >
> > > **Question 3: There isn't a clear motivation for the authors' ranking criterion. While using the uncertainty values as a secondary ordering for data whose hashes collide is certainly sensible, it also seems arbitrary. Could the authors provide a principled justification for why ordering data this way should work better in some cases than something else? Furthermore, one dissatisfactory thing about the authors' method is that the training and the test/retrieval procedures seem to differ. Concretely, the authors use the uncertainty values to extend the hashes of the data in the pool set but don't use them for the query point. Could the authors comment on this?**
> > >
> > > Given a confident pool data sample and an uncertain data sample with the predicted Hash-codes having the same Hamming distance to the query. The uncertain pool data is more likely to be either more relevant to the query, or irrelevant to the query compared to the confident one. However, given the fact that these Hamming distances are small when we are performing nearest neighbor search, it is more likely that the latter one hold true, and we should choose the confident pool data sample to achieve better retrieval accuracy. In a scenario when these Hamming distance are large, the uncertain data will be more likely to be a relevant one.
> > >
> > > The uncertainty value can be indeed used for query data. In Figure 3 and 5, we show that a query point with higher uncertainty will potentially result in higher risk of erroneous retrieval, and the system may warn the user or decide to use a retrieval method with higher accuracy when a query point with high uncertainty is presented.
> > > We have include a new paragraph discuss about this in Section 4.4 of our updated draft.
> > >
> > > **Question 4: The authors below Eq (2) state, "$\lambda$ is a hyperparameter to reflect our belief of the data relative to the prior." I don't understand what this should mean; could the authors please clarify? On a related note, do the authors tune the prior parameters?**
> > >
> > > Adding a scalar balancing the likelihood term and KL term of the Evidence Lower BOund(ELBO) has been discussed in many previous works[2][3] and becomes a practice in some amortized model training for variational inference
> > > [4].
> > > These two terms of the Evidence Lower BOund(ELBO) will represent *model fitness to the training data* and *discrepancy to the prior distribution* respectively.
> > > Multiplying the KL term with a small coefficient will result in the model fitting better to the data, while a large coefficient will lead to the variational posterior close to the prior. Another way to explain it is that multiply the KL term with $\lambda$ will result in the new ELBO to be equivalent to perform Bayesian inference with the new posterior: $p(\boldsymbol{b}|\boldsymbol{x}, y) \propto {p(y|\boldsymbol{x}, \boldsymbol{b})^{\frac{1}{\lambda}}p(\boldsymbol{b}|\boldsymbol{x})}$, which is known as *weighted likelihood*. We do not tune this hyperparameter specifically, but we do perform sensitivity analysis of this $\lambda$ in Section 6.3.2, from which we observe the model achieving satisfying performance as long as the $\lambda$ is set within an appropriate range. We do suggest tuning this hyperparameter when applying our method to other data/scenario.
> > >
> > > [1] Blundell, Charles, et al. "Weight uncertainty in neural network." International conference on machine learning. PMLR, 2015.
> > >
> > > [2] Higgins, Irina, et al. "beta-vae: Learning basic visual concepts with a constrained variational framework." ICLR (Poster) 3 (2017).
> > >
> > > [3] Fu, Hao, et al. "Cyclical annealing schedule: A simple approach to mitigating kl vanishing." arXiv preprint arXiv:1903.10145 (2019).
> > >
> > > [4] Fan, Xinjie, et al. "Bayesian attention modules." Advances in Neural Information Processing Systems 33 (2020): 16362-16376.

---

> ### Comment · Reviewer_Qmeu · 2024-09-09
>
> ## Re: response to my Q1
> I thank the authors for running these additional experiments. I find these results quite surprising; my impression of FFG compared to MCD is that it should provide a much better variational approximation. Did the authors find that FFG exhibited optimisation difficulties, or did they make any non-trivial changes to the optimizer / model architecture?
>
> ## Re: response to my Q2
> The authors' response doesn't seem to address my question/concern; they are just repeating the contents of their paper. To be clear, I don't think using the t-test is necessarily a bad idea. However, besides the fact that the assumptions required for the t-test probably do not hold in HashUQ, I don't see a theoretical justification for it. For example, if the weights were deterministic, is there a connection between using the Shannon entropy vs the t-test?
>
> ## Re: response to my Q3
> I understand the authors' argument, but does experimental evidence back it up, or are they only arguing heuristically?
>
> ## Re: response to my Q4
> Given the response, I understood the method correctly. But please change the sentence I highlighted because, as I quoted it, it doesn't make sense.
>
> I also believe the authors misunderstood my second question. I was interested to know if the authors tune the prior parameters, not $\lambda$.

---

> > ### Author Response · Authors · 2024-09-15
> > **Author Response to Reviewer Qmeu**
> >
> > **Re: response to my Q1
> > I thank the authors for running these additional experiments. I find these results quite surprising; my impression of FFG compared to MCD is that it should provide a much better variational approximation. Did the authors find that FFG exhibited optimisation difficulties, or did they make any non-trivial changes to the optimizer / model architecture?**
> >
> > We did not make any "non-trivial" changes when we implemented the fully factorized Gaussian (FFG) variational inference in our experiments.
> > We did not find any clear empirical evidence showing that FFG can provide better variational approximation than Monte Carlo dropout (MCD) in our setup either. Considering both FFG and MCD as two approximate inference, it is not immediately clear to us why the reviewer would expect that FFG, even if we assume that we indeed find the exact global optimal FFG variational distribution, should definitely perform better than MCD. In the paper that the reviewer provided [1], the main results also showed that:
> >
> > 1. In the *Single Hidden Layer Neural Network*, neither of the fully factorized Gaussian and Monte Carlo dropout can model the ``in-between uncertainty'' well. The fully factorized Gaussian is slightly better than Monte Carlo dropout under this setup.
> >
> > 2. In the *Deeper Network*, both the fully factorized Gaussian and Monte Carlo dropout have the flexibility to model the ``in-between uncertainty'' well. The ELBO optimization objective function might be the reason for a mismatch between variational posterior and true posterior. Notice that this setup also includes the case in which only the weights of the last layer are modeled to be factorized Gaussian.
> >
> > Our setup falls into the second situation: although fully factorized Gaussian can be better in other settings with a shallow structure, we do not find any justification that fully factorized Gaussian should be better than Monte Carlo dropout in our setup with a deep structure, also considering the hierarchical construct of variational distributions for hash-codes in our experiments. In the empirical evaluation from some other works in literature [2], the fully factorized Gaussian does not outperform Monte Carlo dropout as well *in terms of the empirical performance*.
> >
> > *We did not make any non-trivial changes to the optimizer/model architecture for FFG as typically done in literature.* Here we include a more detailed description of our experimental setup and overall procedure to illustrate what we have tested so far:
> >
> > 1. *Implementation:* We tested the fully factorized Gaussian weights with Bayes-by-backpropagation (BBB) implementation (Reparameterization VI and ELBO maximization) [2] using Bayesian-torch [3]. BBB is the most popular implementation of fully factorized Gaussian and has also been adopted in [1].
> >
> > 2. *Training Details and Hyperparameter Settings:* We modeled the weight matrices of the last \{$1, 2, 3$\} fully connected layers as factorized Gaussian and set the learning rates to be \{$1\times 10^{-5}, 5\times 10^{-5}, 1\times 10^{-4}$\}. A standard Gaussian prior was adopted and \{$1, 5, 10$\} Monte Carlo samples were used to reduce the variance of gradient estimation. We tested FFG combined with and without the deterministic dropout. We used the same RMSProp optimizer for both FFG and MCD. The best performing model in all the aforementioned configurations on the ImageNet dataset was chosen to be reported. We have additionally tested other optimizers such as Adam but we did not include it into the grid search of hyperparameters due to the constrained computational resources as well as their limited effects on the empirical performance.
> >
> > 3. *Findings and Discussions:* We found that the empirical performance of FFG is sensitive to two hyperparameters: (1) number of FFG layers, and (2) number of Monte Carlo samples to estimate the ELBO optimization objective. A possible explanation is: Both of a larger number of layers modeled to be Gaussian and a smaller number of Monte Carlo samples will lead to a higher variance of the Monte Carlo estimation of the ELBO optimization objective, which will consequently lead to a high variance of gradient estimation. Moreover, the exponential functions used to ensure the non-negativity of the standard deviations of Gaussian weights can easily get saturated with a diminishing gradient as the values of standard deviations goes smaller and smaller. All the facts listed will lead to a diminishing Signal-to-Noise Ratio, which contributes to the difficulties of gradient-based optimization for FFG with reparameterization VI. In the meanwhile, the empirical finding of the paper [1] suggests the potential problem of using the ELBO optimization objective even with a variational distribution flexible enough.

---

> ### Author Response · Authors · 2024-09-15
> **Author Response to Reviewer Qmeu**
>
> Additionally, a worse empirical result does not necessarily mean a worse approximation to the true posterior considering the existence of potential model mismatch. Given the facts and justifications listed above, we believe that our experiment provides a reasonable comparison between fully factorized Gaussian and Monte Carlo dropout in our framework.
>
> **Re: response to my Q2
> The authors' response doesn't seem to address my question/concern; they are just repeating the contents of their paper. To be clear, I don't think using the t-test is necessarily a bad idea. However, besides the fact that the assumptions required for the t-test probably do not hold in HashUQ, I don't see a theoretical justification for it. For example, if the weights were deterministic, is there a connection between using the Shannon entropy vs the t-test?**
>
> The paired sample $t$-test measures whether the population mean of two groups of samples are equal. *If the weights are deterministic, the $t$-test will not be applicable since there will be no variation in $\pi_{\cdot, k}$.* Here we try to connect the estimated uncertainty by $t$-test statistics to Shannon's entropy based measure of uncertainty. % with the existence of weights randomness instead.
>
> We present an exemplar scenario where the two-sample t-test can be effectively used to rank uncertainty:
>
> - In Case A, for a given input, the posterior samples always predict the probabilities of classes 1 and 2 as $(\pi_{\cdot, k}',\pi_{\cdot, k}'') = (0.3, 0.7)$.
>
> - In Case B, while the average probabilities remain $(0.3, 0.7)$, the values of $(\pi_{\cdot, k}',\pi_{\cdot, k}'')$ fluctuate, ranging from $(0.2, 0.8)$ to $(0.4, 0.6)$.
>
> Clearly, the uncertainty in Case B is greater, and the two-sample t-test captures this difference effectively.
>
> *Paired sample $t$-test based Uncertainty Measure:*
> We conducted the paired sample $t$-test with the null hypothesis $H_0$: $\overline{\pi_{\cdot, k}'} = \overline{\pi_{\cdot, k}''}$. The $t$-statistic of our adopted paired sample test is calculated as follows:
> $$t(\pi_{\cdot, k}', \pi_{\cdot, k}'')  = \frac{\overline{\pi_{\cdot, k}' - \pi_{\cdot, k}''}}{\text{Std}(\pi_{\cdot, k}' - \pi''_{\cdot, k})/\sqrt{N}}$$
>
> where $\overline{\pi_{\cdot, k}' - \pi_{\cdot, k}''}$ and $\text{Std}(\pi_{\cdot, k}' - \pi_{\cdot, k}'')$ represent the sample mean and standard deviation of $(\pi_{n, k}' - \pi''_{n, k})$. $N$ is the number of samples.
>
> The uncertainty of the $k$-th entry is quantified as the tail probability of observing $t(\pi_{\cdot, k}', \pi_{\cdot, k}'')$ under the null hypothesis $H_0$:
> $$pval(t(\pi_{\cdot, k}', \pi_{\cdot, k}'')) = P(t((\tilde{\pi_{\cdot, k}'}, \tilde{\pi_{\cdot, k}''}))\geq t(\pi_{\cdot, k}', {\pi_{\cdot, k}''}) |(\tilde{\pi_{\cdot, k}'}, \tilde{{\pi''_{\cdot, k}}})\sim H_0)$$
>
> *Uncertainty Measured using Conditional Shannon's Entropy:* One way to measure the hash-code uncertainty using Shannon's entropy is to use the conditional Shannon's entropy, which is calculated as follows:
>
> $$\\mathbb{E}\_{\\psi\\sim q(\\psi)}[H(b|\\psi, x)] \approx - \\overline{\\left[\\pi\_{\\cdot, k}'\\log \\pi_{\\cdot, k}' + (1 - \\pi_{\\cdot, k}')\\log (1 - \\pi_{\\cdot, k}')\\right]}.$$
>
> Some previous works use this conditional Shannon's entropy to measure *aleatoric uncertainty*.
>
> *Uncertainty Measured using Total Shannon's Entropy:* Another way to measure the hash-code uncertainty using Shannon's entropy is the following Total Shannon's Entropy:
>
> $$H(b|x) \\approx [\\overline{\\pi'}\_{\\cdot, k}\\log\\overline{\\pi'}_{\\cdot, k}+ (1 - \\overline{\\pi'}\_{\\cdot, k})\\log(1 - \\overline{\\pi'}\_{\\cdot, k})]$$
>
> Some previous works use this total Shannon's entropy to measure *total uncertainty*. An induced *epistemic uncertainty* can be calculated by substracting total Shannon's Entropy to conditional Shannon's entropy:
> $$H(b|x) - \mathbb{E}_{\psi\sim q(\psi)}[H(b|\psi, x)].$$
>
> *Connection and Difference:* Both of the $t$-test and Shannon's entropy give the highest estimation of uncertainty when the model consistently predicts $\pi_{\cdot, k}' \approx 0.5$ (corresponding to high aleatoric uncertainty) while the lowest estimation of uncertainty when the model predicts $\pi_{\cdot, k}' \approx 0$ or $\pi_{\cdot, k}' \approx 1$ (corresponding to low aleatoric uncertainty). However, the uncertainty measured using $t$-test is significantly more sensitive to the variation of $\pi_{\cdot, k}'$ (corresponding to the model uncertainty) compared to Shannon's entropy. When the Shannon's Entropy based measure of uncertainty mentioned above is adopted, the total uncertainty do not change and the aleatoric uncertainty decrease as the variation of $\pi_{\cdot, k}'$ increase. The uncertainty measured using $t$-test will increase when the variation of $\pi_{\cdot, k}'$ increase.

---

> ### Author Response · Authors · 2024-09-15
> **Author Response to Reviewer Qmeu**
>
> We include a numerical comparison between paired sample $t$-test and Shannon's entropy based measures of uncertainty using $100$ samples from truncated Gaussian distributions in Figure 8 and Figure 9 of *Appendix of our updated draft* to show how each of these two uncertainties changes as the mean $\overline{\pi'}$ and variance $\text{Var}(\pi')$ changes.
>
> *Assumption for Paired Samples t-test:* The main assumption for using paired sample $t$-test is that the difference between the paired values is normally distributed. We additionally plot histograms and Q-Q plots of the difference between paired samples of each Bernoulli success rate of hash vectors in Figure 10 of *Appendix of our updated draft*. We observe the differences between paired samples of Bernoulli success rates are approximately normally distributed.
>
> **Re: response to my Q3
> I understand the authors' argument, but does experimental evidence back it up, or are they only arguing heuristically?**
>
> While we designed our strategy in a heuristic way, we have run one more experiment with the most uncertain pool data prioritized and compared to our previously acquired data to support our argument. In the table below, we include the retrieval accuracy of our ProbHash, HashUQ with the confident pool data prioritized (HashUQ-Confident) and HashUQ with the uncertain pool data prioritized (HashUQ-Uncertain) on ImageNet.
>
> | Method    | 16 bit | 32 bit    | 64 bit    |
> |---|---|---|---|
> | ProbHash   | 0.606    | 0.680    | 0.706    |
> | HashUQ-Confident   | 0.631    | 0.696    | 0.718    |
> | HashUQ-Uncertain   | 0.563    | 0.656    | 0.689    |
>
> We observe worse retrieval accuracy of HashUQ-Uncertain compared to HashUQ-Confident and ProbHash. This implies that the uncertain pool data is more likely to be irrelevant to the query compared to not only the confident ones, but also most of the other pool data in between when Hamming distances are small. We are willing to include more experiments if the reviewer can clearly indicate what experiment(s) we should run.
>
> **Re: response to my Q4
> Given the response, I understood the method correctly. But please change the sentence I highlighted because, as I quoted it, it doesn't make sense. I also believe the authors misunderstood my second question. I was interested to know if the authors tune the prior parameters, not $\lambda$.**
>
> We have revised the quoted sentence accordingly in our updated draft.
>
> We did not tune the prior parameters and set the prior to be factorized fair Bernoulli distribution $p(b_{\cdot,k} = 1) = p(b_{\cdot,k} = -1) = 0.5$, which we have mentioned in Appendix Section B.1.2 of our updated draft, for the following reasons:
>
> 1. The fair Bernoulli distribution is the *noninformative prior* and should be used since we do not have the domain knowledge regards the hash-codes distribution.
>
> 2. The factorized fair Bernoulli distribution will promote disentangled binary representations with the best diversity of hash-codes.
>
> [1] Foong, Andrew, et al. "On the expressiveness of approximate inference in Bayesian neural networks." Advances in Neural Information Processing Systems 33 (2020): 15897-15908.
>
> [2] Blundell, Charles, et al. "Weight uncertainty in neural network." International conference on machine learning. PMLR, 2015.
>
> [3] Tran, Dustin, et al. "Bayesian layers: A module for neural network uncertainty." Advances in neural information processing systems 32 (2019).

---

> > ### Comment · Reviewer_Qmeu · 2024-09-18
> >
> > I thank the authors for their detailed reply!
> >
> > Based on the authors' response and the updated manuscript, I no longer have any outstanding concerns regarding the paper and am happy to recommend acceptance.
> >
> > Re: FFG vs MCD: I was surprised because when I have previously worked with variational BNNs, the MCD approximation consistently performed worse compared to FFG on the tasks I was trying to solve, especially in terms of the calibration of the uncertainty estimates. However, the authors have done enough experiments to verify that, in their case, the difference between the methods is not so stark.

---

### Review · Reviewer_79if · 2024-08-19

**Summary Of Contributions:**

The paper proposes a novel image hashing method to improve retrieval tasks by quantifying model uncertainty. The model ranks by both distances between the query image and images in the database and the uncertainty of query images. The model uncertainty is quantified by sampling from a Bayesian network. The proposed method obtains better performance on three datasets of image retrieval.

**Audience:**

Yes

**Broader Impact Concerns:**

No broader impact concerns.

**Claims And Evidence:**

Yes

**Requested Changes:**

Please provide more details of methods and improve writing to clarify the ambiguities.

**Strengths And Weaknesses:**

Strength:

- The paper proposes a novel method that leverages model uncertainty to improve image hashing for retrieval problems. The contribution and the motivation of the method are well justified.

- Sufficient experiments and ablation studies are performed to support the arguments.

Weakness:

- The description of the methodology is not quite clear. The paper proposes two methods ProbHash and HashUQ, but ProbHash is actually a part of HashUQ. It would be better if they were described and compared in the form of ablation instead of two methods.  Also, the logic is not quite coherent. Sections 4 and 5 are titled with "Quantifying the Uncertainty of Hashing" and "Hashing with Uncertainty Quantification", which talk about the same thing but are split into two sections.

- The ambiguities in the paper make it hard to reproduce. In Sec 3.4, how is prior $p(b)$ selected? How does it impact the performance? In Sec 5.1 "$\alpha$ is a positive real number small enough such that...", what is $\alpha$ and how is it set?

- Some general questions about the problem setup - For this image hash problem, will training a classification network and then hash the predicted class better than directly predicting hash codes? Will codes with 4 8-bit classes perform better than 32 binary codes?

- HashUQ requires significantly more computation for inference on the image hash database, making the comparison with the baselines not truly fair.

- Why does Table 1 use bold font on the best model except for full HashUQ?

---

> ### Author Response · Authors · 2024-08-31
> **Author Response to Reviewer 79if**
>
> Thanks for the reviewer for the constructive feedbacks. Here we would like to address the main concerns raised by the reviewer as follows:
>
> **Weakness 1: The description of the methodology is not quite clear. The paper proposes two methods ProbHash and HashUQ, but ProbHash is actually a part of HashUQ. It would be better if they were described and compared in the form of ablation instead of two methods. Also, the logic is not quite coherent. Sections 4 and 5 are titled with "Quantifying the Uncertainty of Hashing" and "Hashing with Uncertainty Quantification", which talk about the same thing but are split into two sections.**
>
> We use *ProbHash* and *HashUQ* to highlight two different concepts:
>
> 1. Probabilistic modeling of hashing functions, with the amortized variational inference.
> 2. Using the quantified uncertainties of hash-codes to improve the retrieval accuracy.
>
> These two concepts are closely related, but mutually exclusive.
> In Section 4, we mainly discuss the *student $t$-test based measure of uncertainty*, which is a new method to quantify the uncertainty for Learning-to-Hash, while in Section 5, we mostly discuss an algorithm to *augment the retrieval using the measured uncertainties* as well as some practical considerations. We appreciate the reviewer for pointing out the potential ambiguity, and we have changed the title of Section 4 to "Measuring the Uncertainty of Hashing" in our updated draft for better clarify.
>
>
> **Weakness2: The ambiguities in the paper make it hard to reproduce. In Sec 3.4, how is prior selected? How does it impact the performance? In Sec 5.1 " is a positive real number small enough such that...", what is and how is it set?**
>
> We apologize for missing some information essential for reproducing the results. We choose fair Bernoulli distribution $p(b_{\cdot,k} = 1) = p(b_{\cdot,k} = -1) = 0.5$ as the prior for each entry of the hash-codes, which is the noninformative prior and should be used since we don't have the domain knowledge regards the hash-codes distribution. The $\alpha$ is introduced as an equivalent way to describe our HashUQ algorithm and a simple way for simulation, and we set $\alpha$ as $\max_{i} | \sum_{k=1}^{K} \log P_{k}(\boldsymbol{x}_{i}^{p}) | $ in all of our simulation. We have included these information in the corresponding section (*Appendix* B.1.2) of our updated manuscripts. We have also submitted the code implementation as the supplementary material to help reproducing our results.
>
> **Weakness3: Some general questions about the problem setup - For this image hash problem, will training a classification network and then hash the predicted class better than directly predicting hash codes? Will codes with 4 8-bit classes perform better than 32 binary codes?**
>
> Training a classification network and then hash the predicted class might achieve better empirical performance in many of the benchmark protocols for image retrieval, but has the following disadvantages for:
>
> 1. Using a classification network and then hash will lead to a loss of non-class information.
> 2. The classification network relies on the pre-defined classes, which might not be applicable for a image which does not belong to the existing classes.
>
> Considering the diversity of the media content from the Internet, preserving some non-class information and being applicable for a image not belonging to the existing classes should be important where the image hashing methods are applied. We acknowledge that these two benefits might not be fully reflected in most of the existing benchmarks for image retrieval, and some new benchmarks which can faithfully reflect the retrieval performance in a real application scenario, for example, with extreme number of classes or ambiguous test images, need to be developed, which might be a future research direction to help establishing retrieval methods.
>
> A combination of 4 8-bit codes might perform better than a 32-bit binary code, but that would require a careful analysis and design of the learning target. Our experimental results show that 4 8-bit codes with ``Label-Target'' construction can give an mAP@1000 of 0.658 on the ImageNet dataset, which is similar to the results of 32-bit binary codes. With careful design of training objectives that can better preserve both of the class and non-class information in each of the 8-bit code, the performance could be further improved potentially.

---

> > ### Author Response · Authors · 2024-08-31
> > **Author Response to Reviewer 79if**
> >
> > **Weakness4 and Weakness5: HashUQ requires significantly more computation for inference on the image hash database, making the comparison with the baselines not truly fair. Why does Table 1 use bold font on the best model except for full HashUQ?**
> >
> > In a scenario where the learning-to-hash is used to perform data retrieval, the hash-codes are usually pre-computed and stored in a database. The retrieval time and storage, rather than inference time, should be considered as the main computational budget. Our proposed uncertainty quantification for hashing will gives a real value confidence, which will take significantly more storage compared to the hash-code itself. We compare baselines, and baselines with two more bits, to HashUQ with binarized uncertainties, which is more fair comparison considering the similar computational and storage consumption in the retrieval phase. The full HashUQ is not compared to the baselines, and only listed as a reference in Table 1. We acknowledge the extra inference time is a potential problem of our method, which could be potentially solved by training a student model to distill the information from the approximated Bayesian neural network.

---

### Author Response · Authors · 2024-08-31
**Updated Manuscript Available for Review**

We would like to thank the reviewer **3tbV, Qmeu, 79if** for the valuable suggestions. We are pleased that the reviewers recognize the novelty of this work and the strength in experimental evaluation. We try our best to further revise our manuscript to reflect the suggestions and address the concerns from the reviewers. We mark our revisions to the updated manuscript in blue, and here is a summary of the major revisions:

1. We have changed the organization of Section 2.4 to present two variant hashing model constructs of our method separately for clearer presentation. We have also added one more paragraph to discuss the connection to variational information bottleneck.
2. We have moved the literature review section to the end of the paper and added one more paragraph to discuss differences and novelty of our work compared to existing works.
3. We have moved the definition of mean average precision from the *Appendix* to Section 5.1 of the *Main Text* and added one more section to define the Hadamard matrix in *Appendix* A.1.
4. We have added new experimental results to empirically compare MC Dropout to Fully Factorized Gaussian in the newly added section, *Appendix*B.5, as the support for our model choice.
5. We have changed the notations to a double subscript to avoid clashing as suggested by reviewer **Qmeu**.

---

### Decision · Action_Editor_hmNb · 2024-09-22

**Recommendation:** Accept with minor revision

**Comment:**

This paper studies using neural networks to model hash functions for uncertainty-aware image retrieval. The proposed framework, HashUQ, first predicts codes and uncertainty estimations for input images in the encoding stages. The retrieval process can then leverage the uncertainties for better ranking results. The reviewers agree that using uncertainty estimation to augment information retrieval is a strong idea (79if, Qmeu), and the experiment mostly shows its effectiveness (all reviewers). Meanwhile, reviewers also raise several concerns regarding result comparisons with specific baselines (Qmeu), motivation of some algorithm designs (Qmeu, 3tbv), and clarities & presentation issues (79if, Qmeu). The author makes a good effort in the rebuttal phase and addresses most of the concerns. Additionally, reviewer 3tbv also requests an analysis of the likelihood distribution assumption made in the center-target method. The AE checked the discussion threads and revised paper, finding that the authors include a discussion in Appendix A.3 of the latest revised version. Given the consensus of all reviewers,  the AE recommends the acceptance of this paper, with the request of performing an experimental comparison between the selected Boltzmann distribution versus other commonly used distributions, as requested by reviewer 3tbv.

**Audience:**

Yes.

**Claims And Evidence:**

Yes.